# ODEFormer: Symbolic Regression of Dynamical Systems with Transformers

**Stéphane d'Ascoli**[*]
EPFL
stephane.dascoli@gmail.com

**Sören Becker**[*]
Helmholtz Munich
Munich Center for Machine Learning
TU Munich
soren.a.becker@gmail.com

**Alexander Mathis**
EPFL

**Philippe Schwaller**
EPFL

**Niki Kilbertus**
Helmholtz Munich
Munich Center for Machine Learning
TU Munich

## Abstract

We introduce ODEFormer, the first transformer able to infer multidimensional ordinary differential equation (ODE) systems in symbolic form from the observation of a single solution trajectory. We perform extensive evaluations on two datasets: (i) the existing 'Strogatz' dataset featuring two-dimensional systems; (ii) ODEBench, a collection of one- to four-dimensional systems that we carefully curated from the literature to provide a more holistic benchmark. ODEFormer consistently outperforms existing methods while displaying substantially improved robustness to noisy and irregularly sampled observations, as well as faster inference. We release our code, model and benchmark at https://github.com/sdascoli/odeformer.

## 1 Introduction

Recent triumphs of machine learning (ML) spark growing enthusiasm for accelerating scientific discovery (Davies et al., 2021; Jumper et al., 2021; Degrave et al., 2022). In particular, inferring dynamical laws governing observational data is an extremely challenging task that is anticipated to benefit substantially from modern ML methods. Modeling dynamical systems for forecasting, control, and system identification has been studied by various communities within ML. Successful modern approaches are primarily based on advances in deep learning, such as neural ordinary differential equation (NODE) (see Chen et al. (2018) and many extensions thereof). However, these models typically lack interpretability due to their black-box nature, which has inspired extensive research on explainable ML of overparameterized models (Tjoa & Guan, 2020; Dwivedi et al., 2023). An alternative approach is to directly infer human-readable representations of dynamical laws.

This is the main goal of symbolic regression (SR) techniques, which make predictions in the form of explicit symbolic mathematical expressions directly from observations. Recent advances in SR make it a promising alternative to infer natural laws from data and have catalyzed initial successes in accelerating scientific discoveries (Aréchiga et al., 2021; Udrescu & Tegmark, 2021; Butter et al., 2021). So far, SR has most commonly been used to infer a function $g(x)$ from paired observations $(x, g(x))$ – we call this *functional SR*. However, in many fields of science, understanding a system involves deciphering its dynamics, i.e., inferring a function $f(x)$ governing its evolution via an ODE $\dot{x} = f(x)$ – we call this setting *dynamical SR*. The task is then to uncover $f$ from an observed solution trajectory $(t, x(t))$, where observations of $x(t)$ may be noisy and times $t$ may be irregularly sampled.

**Contributions.** In this work, we introduce ODEFormer, the first Transformer trained to infer dynamical laws in the form of multidimensional ODEs from observational (noisy, irregularly sampled) data. It relies on large-scale training of a sequence-to-sequence transformer on diverse synthetic data, leading to efficient and scalable inference for unseen trajectories. Faced with the lack of benchmarks

---

[*]Equal contribution.

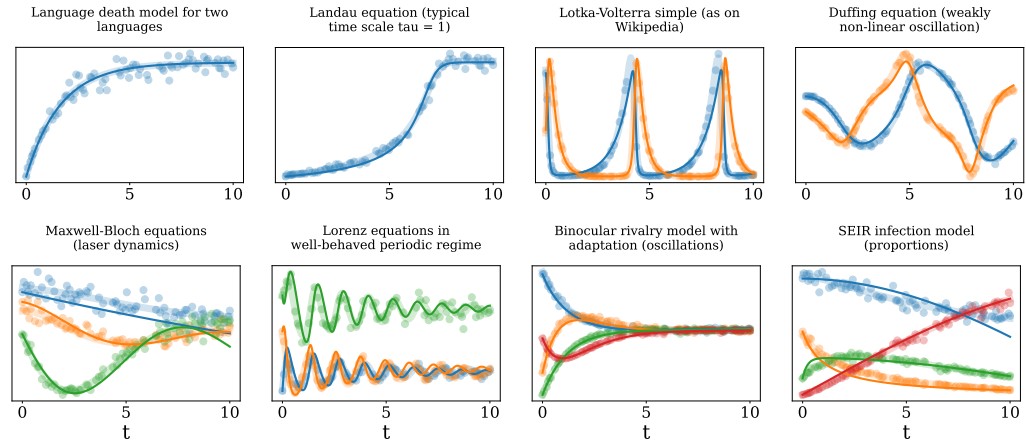

Figure 1: **ODEFormer predicts dynamics from noisy and irregularly sampled trajectories in 1-4 dimensional systems.** Ground truth trajectories (thick lines) are corrupted (5% noise) and unevenly sampled (50% of the equally spaced points are dropped uniformly at random), yielding the observed data (dots) from which ODEFormer infers a symbolic ODE. Integrating the predicted ODE from the original initial values gives a trajectory (thin lines) that closely approximates the observed data.

for dynamical SR (the only existing one, called "Strogatz dataset" (La Cava et al., 2021), contains only seven two-dimensional systems, and is not integrated with sufficient precision (Omejc et al., 2023)), we also introduce ODEBench, a more extensive dataset of 63 ODEs curated from the literature, which model real-world phenomena in dimensions one to four (see Appendix A for details). On both benchmarks, ODEFormer achieves higher accuracy than existing methods for irregular, noisy observations while being faster than most competitors.

**Problem setting and overview.** We assume observations $\{(t_i, x_i)\}_{i \in [N]}$, of a solution $x(t)$ of $\dot{x} = f(x)$, for some $f : \mathbb{R}^D \to \mathbb{R}^D$, where $x_i$ can be noisy observations of $x(t_i)$ with irregularly sampled $t_i$. Here, we use the notation $[N] := \{1, \ldots, N\}$ and $\dot{x}$ for the temporal derivative. The task is to infer $f$ in symbolic form from $\{(t_i, x_i)\}_{i \in [N]}$. As illustrated in Figure 2, ODEFormer is based on large-scale pre-training on "labelled" examples that consist of randomly generated symbolic mathematical expressions as the prediction target $f$ and a discrete solution trajectory $\{(t_i, x_i)\}_{i \in [N]}$ as input, obtained by integrating $\dot{x} = f(x)$ for a random initial condition.

## 2 RELATED WORK

**Modeling dynamical systems.** We briefly mention some relevant cornerstones from the long history of modeling dynamical systems from data, each of which has inspired a large body of follow-up work. Neural ODEs (NODE) (Chen et al., 2018) parameterize the ODE $f$ by a neural network and train it by backpropagating through ODE solvers (either directly or via adjoint sensitivity methods). NODEs require no domain knowledge but only represent the dynamics as a generally overparameterized black-box model void of interpretability. Assuming prior knowledge about the observed data, physics-informed neural networks (Raissi et al., 2019; Karniadakis et al., 2021) aim to model dynamical systems using neural networks regularized to satisfy a set of physical constraints. This approach was recently extended to uncover unknown terms in differential equations (Podina et al., 2023). In this work, we aim to infer interpretable dynamical equations when no domain knowledge is available.

**Approaches to symbolic regression.** While it is possible to generate complex analytical expressions that fit the observations, an unnecessarily lengthy function is often impractical. Symbolic regression seeks a balance between fidelity to the data and simplicity of form. Therefore, predictions are typically also evaluated in terms of some "complexity" metric. Because the symbolic output makes it difficult to formulate differentiable losses, SR has traditionally benefitted comparably little from advances in autodifferentiation and gradient-based optimization frameworks. The dominant approach has thus been based on evolutionary algorithms such as genetic programming (GP) (La Cava et al., 2016b; 2018; Kommenda et al., 2020; Virgolin et al., 2021; Tohme et al., 2022; Cranmer, 2023), optionally guided by neural networks (Mundhenk et al., 2021; Udrescu & Tegmark, 2020; Costa et al., 2021) and recently also employing reinforcement learning (Petersen et al., 2020) – see (La Cava et al., 2021; Makke & Chawla, 2022) for reviews. Most of these approaches require a separate optimization for each new observed system, severely limiting scalability.

Figure 2: **Model overview**. ODEFormer maps (noisy) observations to symbolic sequences that represent (the right-hand side of) an ordinary differential equation (ODE). Model input includes representations of all state variables (blue & orange) and time (gray). The model is optimized via cross-entropy loss between the softmax distribution over the model vocabulary (gray) vs. the one-hot encoded true target symbol (green). Target sequences are predicted in prefix notation. The illustration includes the first three predictions $[+, \ x_2, \ x_1]$ for the target sequence $[+, \ x_1, \ x_2]$. This example showcases that the loss may be high despite mathematical equivalence between predicted and target sequence - this does not impact training in practice.

**Transformers for symbolic regression.** With the advent of transformer models (Vaswani et al., 2017), efficient learning of sequence-to-sequence tasks for a broad variety of modalities has become feasible. Paired with large-scale pre-training on synthetic data, transformers have been used for symbolic tasks such as integration (Lample & Charton, 2019), formal logic (Hahn et al., 2021), and theorem proving (Polu & Sutskever, 2020). Few recent works applied them to functional SR (Biggio et al., 2021; Valipour et al., 2021; Kamienny et al., 2022; Vastl et al., 2022) obtaining comparable results to GP methods, with a key advantage: after one-time pre-training, inference is often orders of magnitude faster since no training is needed for previously unseen systems. Landajuela et al. (2022) recently proposed a hybrid system combining and leveraging the advantages of most previous approaches for state of the art performance on functional SR tasks.

**Symbolic regression for dynamical systems.** In principle, dynamical SR, inferring $f$ from $(t, x(t))$, can be framed as functional SR for $(x(t), \dot{x}(t))$ pairs. However, when transitioning from functional to dynamical symbolic regression, a key challenge is the absence of regression targets $\dot{x}(t)$ since temporal derivatives are usually not observed directly. A common remedy is to use numerical approximations of the missing derivatives as surrogate targets instead. This approach has been employed in the GP community (Gaucel et al., 2014; La Cava et al., 2016a; Kronberger et al., 2020; Quade et al., 2016) and is also key to the widely used SINDy (Brunton et al., 2016) algorithm which performs sparse linear regression on a manually pre-defined set of basis functions. While SINDy is computationally efficient, its modeling capacity is limited to linear combinations of its basis functions. Like in functional SR, neural networks have also been combined with GP for dynamical SR (Atkinson et al., 2019), and the divide-and-conquer strategy by Udrescu & Tegmark (2020) has also been extended to inferring dynamical systems (Weilbach et al., 2021). Omejc et al. (2023) recently introduced ProGED, which performs dynamical SR via random search of candidate equations, constrained by probabilistic context-free grammars (PCFGs). However, this approach requires knowledge about the ground truth as parameters of the PCFGs need to be carefully tailored to each problem. Finally, Qian et al. (2021) recently presented an innovative loss function to mitigate finite difference approximations which are error-prone in the presence of signal corruptions. In the realm of transformers, d'Ascoli et al. (2022) infer one-dimensional recurrence relations in sequences of numbers, which as discrete maps are closely related to differential equations. Most related to our work, Becker et al. (2023) explore a transformer-based approach to dynamical SR for ODEs (for a detailed comparison, see Appendix H). However, their method is limited to univariate ODEs. Such systems exhibit extremely limited behavior, where solution trajectories can only either monotonically diverge or monotonically approach a fixed value – not even inflections let alone oscillations are possible (Strogatz, 2000). In this work, we tackle the important yet unsolved task of efficiently discovering arbitrary non-linear ODE systems in symbolic form directly from data in multiple dimensions, without assuming prior knowledge of the ground truth.

**Theoretical identifiability.** Traditionally, the identifiability of ODEs from data has been studied in a case-by-case analysis for pre-defined parametric functions (Åström & Eykhoff, 1971; Miao et al., 2011; Villaverde et al., 2016; Hamelin et al., 2020). Recent work made progress on identifiability within function classes such as linear (in parameters) autonomous ODE systems (Stanhope et al., 2014; Duan et al., 2020; Qiu et al., 2022) or, in the scalar case, even non-parametric classes such as analytic, algebraic, continuous, or smooth functions (Scholl et al., 2023). As it stands, it is difficult to gain practical insights from these results for our setting as they do not conclude whether non-linear ODEs can practically be uniquely inferred from data – see Appendix B for details.

## 3    DATA GENERATION

Our method builds on pretraining on a large dataset of ODEs which is assembled as follows.

**Generating ODEs.**   For a $D$-dimensional ODE $f$, we independently sample the $D$ component functions $f_1, \ldots, f_D$ as random unary-binary trees following the method of (Lample & Charton, 2019), where internal nodes represent mathematical operators and leaves represent constants or variables. In our specific procedure, we first sample the system dimension $D$ uniformly from $[D_{\max}]$ for a fixed $D_{\max} \in \mathbb{N}$ and then perform the following steps for each component function:

1. Sample the number of binary operators $b$ uniformly from $[b_{\max}]$ for a fixed $b_{\max} \in \mathbb{N}$.
2. Sample a binary tree with $b$ non-leaf nodes, following the procedure of  Lample & Charton (2019).
3. Decorate each non-leaf node with a binary operator sampled from $P(+) = 3/4$ and $P(\times) = 1/4$.[1]
4. For each leaf in the tree, sample one of the variables $x_i$ for $i \in [D]$.
5. Sample the number of unary operators $u$ uniformly from $[u_{\max}]$ for a fixed $u_{\max} \in \mathbb{N}$.
6. Iteratively with $u$ repetitions, select a node whose subtree has a depth smaller than $6$[2] and insert a new node directly above. Populate the new node with a unary operator that is sampled uniformly at random from $\{x \mapsto \sin(x), x \mapsto x^{-1}, x \mapsto x^2\}$.
7. Convert the tree into a mathematical expression via preorder traversal (Lample & Charton, 2019).
8. Finally, prepend a coefficient to each term and wrap the argument of any unary operator in an affine transformation $x \mapsto a \cdot x + b$. Coefficients and constants of affine transformations are sampled independently from a log-uniform distribution on $[c_{\min}, c_{\max}]$ with $c_{\min}, c_{\max} \in \mathbb{R}$.

Due to random continuous constants (and initial conditions), we almost surely never sample a function twice. In our experiments, we use $D_{\max} = 6$, $b_{\max} = 5$, $u_{\max} = 3$, $(c_{\min}, c_{\max}) = (0.05, 20)$.

**Integrating ODEs.**   Once the function $f$ is generated, we sample an initial condition $x_0 \sim \mathcal{N}(0, \gamma \mathbb{I}_D)$ for a fixed $\gamma \in \mathbb{R}_{>0}$ and identity matrix $\mathbb{I}_D$, and integrate the ODE from $t = 1$ to $t = T$ using the numerical solver `scipy.integrate.solve_ivp` provided by SciPy (Virtanen et al., 2020). The `solve_ivp` function defaults to an adaptive 5th order explicit Runge Kutta method with 4th order error control (Dormand & Prince, 1980) as well as relative and absolute tolerances of $10^{-3}$ and $10^{-6}$ respectively. To avoid the sampling grid to be informative of signal variation, we use the solver's internal option to interpolate the adaptive step-size solution to obtain the final solution on a equidistant grid of $N$ sampling points, where $N \sim \mathcal{U}\{50, 200\}$. When integration fails, i.e., when the solver throws an error, returns unsuccessfully, or takes longer than one second, we simply discard the current sample. Section 4 explains in detail that we can fix $T$ and $\gamma$ during training without loss of generality, as we can rescale observations at inference time. Hence, we use $\gamma = 1$ and $T = 10$.

**Filtering data.**   Under the distribution over functions $f$ defined implicitly by our generator, a substantial fraction of sampled ODEs (and initial conditions) leads to solutions where at least one component diverges over time. Continued divergence over long time spans is typically deemed "unphysical". Among the trajectories that remain finite, we observe that again a substantial fraction swiftly converges towards a fixed point. Although these solutions may be realistic, their dominance hampers diversity in our dataset. Also, stable constant observations over long times spans are arguably rarely of interest. Hence, we use the following heuristics to increase diversity of the generated ODEs:

- If any variable of the solution trajectory exceeds a fixed threshold ($10^2$ in our experiments), we discard the example. This amounts to filtering out divergent systems.
- If the oscillation of all component functions over the last quarter of the integration range is below a certain threshold ($10^{-3}$ in our experiments), we discard the example with a probability of 90%.[3] This filters out a majority of rapidly converging systems.

## 4    MODEL, TRAINING, AND INFERENCE

ODEFormer is an encoder-decoder transformer  (Vaswani et al., 2017) for end-to-end dynamical SR as illustrated in Figure 2. The model comprises 16 attention heads and an embedding dimension

---

[1]Subtractions and divisions are included via multiplication with negative numbers and the unary operator $x \mapsto x^{-1}$ respectively. It has been argued that divisions appear less frequently than additions and multiplications in "typical" expressions (Guimerà et al., 2020).

[2]This aims at avoiding deeply nested uninterpretable expressions, which often occur in GP-based SR.

[3]The oscillation of a function $h : [a, b] \to \mathbb{R}$ is given by $\sup_{x \in [a,b]} h(x) - \inf_{x \in [a,b]} h(x)$.

of 512, leading to a total parameter count of 86M. As observed by Charton (2022), we find that optimal performance is achieved in an asymmetric architecture, using 4 layers in the encoder and 16 in the decoder. As measurement time is explicitly included in the inputs, we remove positional embeddings from the encoder. ODEFormer is trained on the data set of 50M samples generated as described in Section 3 following established principles, with details given in Appendix C. Whereas the transformer itself follows the standard architecture, we adapted tokenization and embedding to the particular task, i.e., inference of symbolic functions from multivariate observations.

**Tokenizing numbers.** Since numeric input trajectories as well as symbolic target sequences may contain floating point values, we need an efficient encoding scheme that allows the infinite number of floats to be sufficiently well conserved by a fixed-size vocabulary. Following Charton (2022), each number is rounded to four significant digits and disassembled into three components: sign, mantissa and exponent, each of which is represented by its own token. This tokenization scheme condenses the vocabulary size required to represent floating point values to just 10203 tokens (+, -, 0, ..., 9999, E-100, ..., E100) and works well in practice despite the inherent loss of precision. We also experimented with three alternative representations: (i) two-token encoding, where the sign and mantissa are merged together, (ii) one-token encoding where sign, mantissa and exponent are all merged together, (iii) a two-hot encoding inspired by Schrittwieser et al. (2020) and used by Becker et al. (2023), which interpolates linearly between fixed, pre-set values to represent continuous values. These representations have the advantage of decreasing sequence length, and (iii) has the added benefit of increased numerical precision for the inputs. Since all three alternatives led to worse overall performance, we used the three token representation (sign, mantissa, exponent).

**Embedding observed trajectories.** For a $D$ dimensional ODE system, the above tokenization scheme represents data of a single observed time-point $(t_i, \mathbf{x_i})$ using $\mathbb{R}^{(D+1)} \times 3$ tokens. Each such token is fed separately to an embedding layer and the resulting token embeddings are concatenated to give a combined embedding vector in $\mathbb{R}^{((D+1) \times 3) \times d_{\text{emb}}}$ for each observed time-point $i$. We want a single ODEFormer model to flexibly handle systems of different dimensionalities (up to $D_{\max} = 6$). In case the observed system has dimension $D < D_{\max}$ we hence fill vacant dimensions via zero-padding. Following Kamienny et al. (2022), the resulting embedding vector are subsequently processed by a 2-layer fully-connected feed-forward network (FFN) with Sigmoid-weighted linear unit (SiLU) activation functions (Elfwing et al., 2018), which reduces the dimension of the final representation of a single time point to $d_{\text{emb}}$ dimensions. This embedding process allows a single trained model to flexibly handle input trajectories of variable lengths and ODE system dimensions.

**Encoding symbolic functions.** In addition to the tokens used to represent floating point values as described above, the vocabulary of the decoder includes specific tokens for all operators and variables. Importantly, the decoder is trained on expressions in prefix notations to relieve the model from predicting parentheses (Lample & Charton, 2019). For $D$-dimensional systems, we simply concatenate the encodings of the $D$ component functions, separated by a special token " | ". With these choices, the target sequence for an exemplary 2-dimensional ODE given by component functions $f_1(\mathbf{x}) = \cos(2.4242 \cdot x_1) + x_2$ and $f_2(\mathbf{x}) = x_1 \cdot x_2$ corresponds to the following token sequence `[+ cos mul + 2424 E-3 x_1 x_2 | mul x_1 x_2]`. Using this simple method, the sequence length scales linearly with the dimensionality of the system, i.e., the number of variables. While this is unproblematic for small dimensions such as $D \leq D_{\max} = 6$, it may impair the scalability of our approach.[4] As the encoder is only concerned with numeric input trajectories, its vocabulary only includes tokens for numbers.

**Decoding strategy.** At inference, we use beam sampling (Van Gael et al., 2008) to decode candidate equations, and select the candidate with highest reconstruction $R^2$ score.[5] For this, beam temperature is an important parameter to control diversity – as the beam size increases, it typically becomes useful to also increase the temperature to maintain diversity. Unless stated otherwise, we perform our experiments with a beam size of 50 and a temperature of 0.1.

**Inference-time rescaling.** During training, the model only observes trajectories integrated on the interval $[1, 10]$ and starting from an initial condition sampled according to a standard normal

---

[4]A possible alternative would be to treat the decoding of each component function as a separate problem, adding a specifier to the BOS (beginning of sequence) token to identify which component is to be decoded.

[5]Beam search tends to produce candidates which all have the same skeleton, and only differ by small variations of the constants, leading to a lack of diversity. Beam sampling ensures that randomness is added at each step of decoding leading to a more diverse set of candidate expressions.

distribution, as described in Section 3. To accommodate for different scales of initial conditions and time ranges encountered during inference, we apply the affine transformations $t \mapsto \tilde{t} = at + b$ to rescale the observed time range to $[1, 10]$ and $x_i(t) \mapsto \tilde{x}_i(t) = x_i(t)/x_i(t_0)$ to rescale initial values to unity. The prediction $\tilde{f}$ that ODEFormer provides on inputs $(\tilde{t}, \tilde{x})$ are then transformed as $f_i = \frac{dx_i}{dt} = \frac{1}{ax_i(t_0)} \frac{d\tilde{x}_i}{d\tilde{t}} = \frac{1}{ax_i(t_0)} \tilde{f}_i$ to recover original units.

**Optional parameter optimization.** Most SR methods break the task into two subroutines (possibly alternating between the two): predicting the optimal "skeleton", i.e., equation structure, and fitting of the constants contained in the skeleton. Just like Kamienny et al. (2022), our model is end-to-end, in the sense that it handles both subroutines simultaneously. However, we also experimented with adding an extra parameter optimization step, as performed in methods such as ProGED (Omejc et al., 2023). We describe the details of the parameter optimization procedure in Appendix D and denote this method as "ODEFormer (opt)".

## 5   EXPERIMENTS

**Experimental tasks.** For dynamical SR there is a spectrum of reasonable tasks. We could simply be interested in finding *some* ODE $\hat{f}$, whose solution approximates the observed trajectory $x(t)$ on the observed time interval, even if $\hat{f}$ still differs from $f$ on (parts of) their domain. A more ambitious goal closer to full identification is to find a $\hat{f}$ that also approximates the correct trajectories for unobserved initial conditions. This evaluation, which in our view is most meaningful to assess dynamical SR, is often absent in the literature, e.g., from Omejc et al. (2023). If an inferred $\hat{f}$ yields correct solutions for all initial values and time spans, we would consider it as perfect identification of $f$. Accordingly, we evaluate the following aspects of performance in our experiments.

- *Reconstruction:* we compare the (noiseless, dense) ground truth trajectory with the trajectory obtained by integrating the predicted ODE from the same initial condition and on the same interval $[1, T]$ as the ground truth.
- *Generalization:* we integrate both the ground truth and the inferred ODE for a new, different initial condition over the same interval $[1, T]$ and compare the obtained trajectories.

**Metrics.** When evaluating SR methods, the desired metric is whether the inferred ODE $\hat{f}$ perfectly agrees symbolically with the ground truth expression $f$. However, such an evaluation is problematic primarily because (i) the ambiguity in representing mathematical expressions and non-determinism of "`simplify`" operations in modern computer algebra systems render comparisons on the symbolic level difficult, and (ii) expressions may differ symbolically, while evaluating to essentially the same outputs on all possible inputs (e.g., in the presence of a negligible extra term). Hence, comparing expressions numerically on a relevant range is more reliable and meaningful. Even modern computer algebra systems include numerical evaluations in equality checks, which still require choosing a range on which to compare expressions (Meurer et al., 2017). We consider the following metrics.

- *Accuracy:* A classical performance metric for regression tasks is the coefficient of determination, defined as $R^2 = 1 - \sum_i (y_i - \hat{y}_i)^2 / \sum_i (y_i - \bar{y})^2 \in (-\infty, 1]$. Since $R^2$ is unbounded from below, average $R^2$-scores across multiple predictions may be severely biased by even a single particularly poor outlier. We therefore instead report the percentage of predictions for which the $R^2$ exceeds a threshold of 0.9 in the main text and show the distribution of scores in Appendix I.
- *Complexity:* we define the complexity of a symbolic expression to be the total number of operators, variables and constants. We acknowledge that this is a crude measure, for example, it would assign a lower complexity to $\exp(\tan(x))$ (complexity=3) than to $1 + 2x$ (complexity=5). Yet, there is no agreed-upon meaning of the term "complexity" for a symbolic expression in this context so that simply counting "constituents" is common in the literature, see e.g. the discussion in the benchmark by La Cava et al. (2021).
- *Inference time:* time to produce a prediction.

**Datasets.** We evaluate ODEFormer on a test set which is generated following the description in Section 3. Although the test set is sampled from the same distribution as the training data, the presence of continuous constants within every sample (step 8 in the sampling procedure in Section 3) as well as i.i.d. initial values ensures that the test data does not overlap with the training data. In addition we also evaluate our approach on two datasets not obtained via our generation procedure.

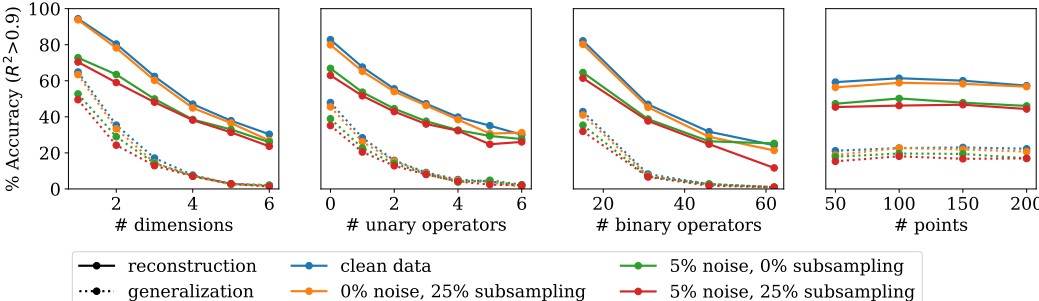

Figure 3: **Ablation study on synthetic data.** We vary four parameters governing the difficulty of an example (from left to right): the dimension of the system, the total number of unary and binary operators, and the number of points in the trajectory. In each panel, we average the results over a dataset of 10,000 examples.

For this we first consider the "**Strogatz dataset**", included in the Penn Machine Learning Benchmark (PMLB) database (La Cava et al., 2021). It consists of seven ODE systems and has been used as a benchmark by various SR methods, in particular those specialized on dynamical SR (Omejc et al., 2023). However, it has several limitations: (i) it is small (only seven unique ODEs, each integrated for 4 different initial conditions), (ii) it only contains 2-dimensional systems, (iii) it is not integrated with sufficient precision (Omejc et al., 2023), and (iv) its annotations are misleading (e.g., claiming that all systems develop chaos even though none of them does). Faced with these limitations, we introduce **ODEBench**, an extended benchmark curated from ODEs that have been used by Strogatz (2000) to model real-world phenomena as well as well-known systems from Wikipedia. We fix parameter values to obtain the behavior the models were developed for, and choose two initial conditions for each equation to evaluate generalization. ODEBench consists of 63 ODEs (1D: 23, 2D: 28, 3D: 10, 4D: 2), four of which exhibit chaotic behavior. We publicly release ODEBench with descriptions, sources of all equations, and well integrated solution trajectories – more details are in Appendix A.

**Corruptions.** For all datasets, we also compare models on their robustness to two types of corruption: (i) to simulate imprecise observations we add measurements noise to the observations via $x_j(t_i) \to (1 + \xi)x_j(t_i)$ for $j \in [D], i \in [N]$ and $\xi \sim \mathcal{N}(0, \sigma)$; (ii) to simulate missing data and imperfect sampling processes, we drop a fraction $\rho$ of the observations along the trajectory uniformly at random. We report results for various noise levels $\sigma$ and subsampling fractions $\rho$. In addition, an experiment when entire intervals of data are missing (e.g. due to a sensor fault) can be found in Appendix F.

**Baselines.** In our experiments, we extensively compare ODEFormer with (strong representatives of) existing methods described in Table 1. All baseline models are fitted to each equation separately, including a separate hyperparameter optimization per equation. In contrast, we evaluate a single, pre-trained ODEFormer model on all equations in all benchmarks. ODEFormer is hence the only model which does not require any hyperparameter tuning or prior knowledge on the set of operators to be used. Apart from ProGED and SINDy, all baselines were developed for functional SR. We use them for dynamical SR as described in Section 2, by computing temporal derivatives $\dot{x}_i(t)$ via finite differences with hyperparameter search on the approximation order and optional use of a Savitzky-Savgol filter for smoothing. For more details, please refer to Appendix E. Moreover, for a performance comparison to the closely related univariate ODE inference model by Becker et al. (2023) see Appendix H.2.

## 5.1 RESULTS

**Synthetic data** We first assess how the performance of ODEFormer is affected by the dimensionality of the ODE system, the number of operators, and the number of points in the trajectory. Results on a test set comprising 10,000 i.i.d. examples generated as described in Section 3 are shown in Figure 3 – we asses the effect of the beam size in Appendix G. We make three observations:

- Performance degrades with the system dimensionality as well as number of unary and binary operators as expected while ODEFormer is surprisingly insensitive to the number of points in the trajectory across the tested range.
- Generalization accuracy is substantially lower than reconstruction accuracy as expected, but at least for low-dimensional systems we achieve non-trivial generalization (e.g., 60% generalization accuracy vs 85% reconstruction accuracy for 1D).
- ODEFormer copes well with subsampling, but suffers more from noisy trajectories. However, the effect on generalization is smaller than that on reconstruction.

Table 1: **Overview of models.** f.d.: finite differences required, ode: method developed for dynamical SR, T: transformer-based, GP: genetic programming, MC: Monte Carlo, reg: regression

| name | type | ode | f.d. | description | reference |
|------|------|-----|------|-------------|-----------|
| ODEFormer | T | yes | no | seq.-to-seq. translation | ours |
| AFP | GP | no | yes | age-fitness Pareto optimization | (Schmidt & Lipson, 2011) |
| FE-AFP | GP | no | yes | AFP with co-evolved fitness estimates | (Schmidt & Lipson, 2011) |
| EHC | GP | no | yes | AFP with epigenetic hillclimbing | (La Cava, 2016) |
| EPLEX | GP | no | yes | epsilon-lexicase selection | (La Cava et al., 2016b) |
| PySR | GP | no | yes | AutoML-Zero + simulated annealing | (Cranmer, 2023) |
| SINDy | reg | yes | yes | sparse linear regression | (Brunton et al., 2016) |
| FFX | reg | no | yes | pathwise regularized ElasticNet regression | (McConaghy, 2011) |
| ProGED | MC | yes | yes | MC on probabilistic context free grammars | (Omejc et al., 2023) |

**Reconstruction results.** We present results on both "Strogatz" and ODEBench in Figure 4. From top to bottom, we ranked methods by their average accuracy across all noise and subsampling levels. The ranking is similar on the two benchmarks and ODEFormer achieves the highest average score on both. The two leftmost panels show that ODEFormer is only occasionally outperformed by PySR when the data is very clean – as noise and subsampling kick in, ODEFormer gains an increasingly large advantage over all other methods. In the two rightmost panels of Figure 4, we show the distributions of complexity and inference time. ODEFormer runs on the order of seconds, versus minutes for all other methods except SINDy, while maintaining relatively low and consistent equation complexity even at high noise levels. We show figures for all predictions in Appendix A.

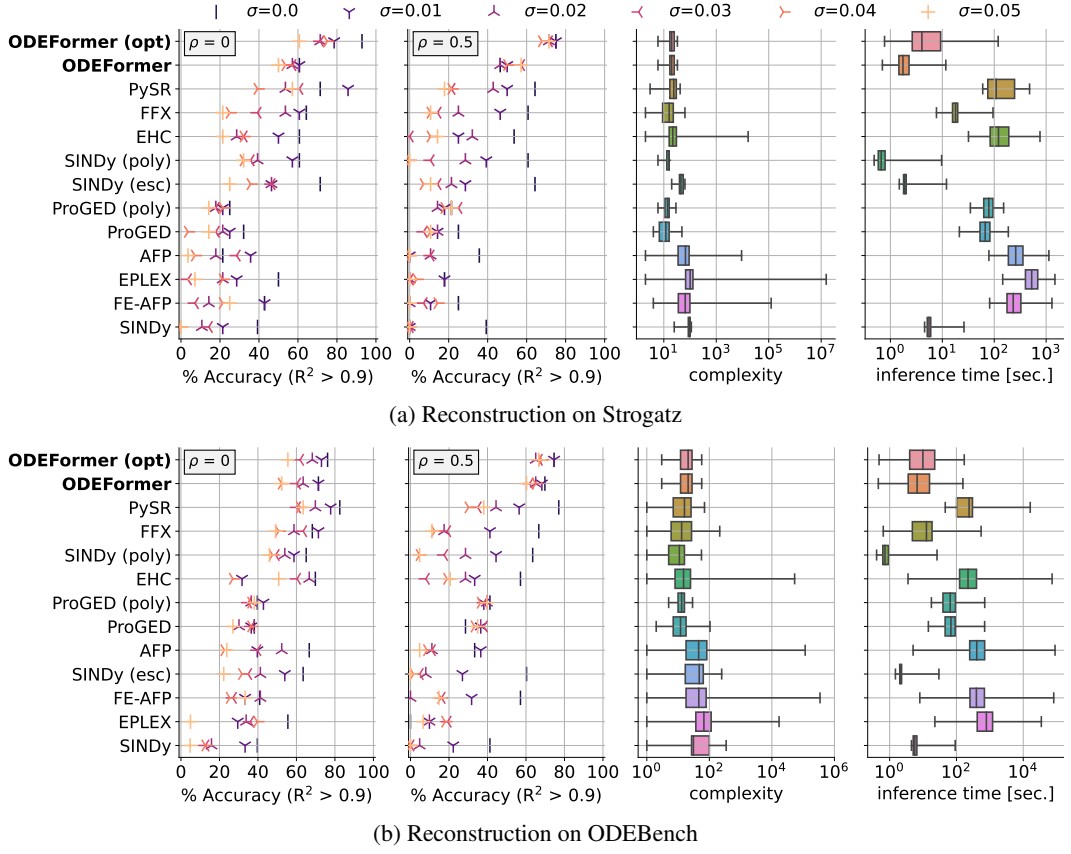

(a) Reconstruction on Strogatz

(b) Reconstruction on ODEBench

Figure 4: **Performance comparison.** We compare ODEFormer (and ODEFormer (opt) for additional parameter optimization) to existing methods, following the protocol described in Section 5. We present results for two values of the subsampling parameter $\rho$ and six values of the noise parameter $\sigma$. Whiskers in box plot panels mark minimum and maximum values.

**Generalization results.** We present generalization results on ODEBench in Figure 5. Consistently across all models, accuracies drop by about half, meaning that half the correctly reconstructed ODEs do not match the ground truth symbolically. This highlights the importance of evaluating dynamical SR on different initial conditions. Note, however, that the overall rankings of the different methods is rather consistent with the reconstruction results, and ODEFormer achieves the best results on average thanks to its robustness to noise and subsampling.

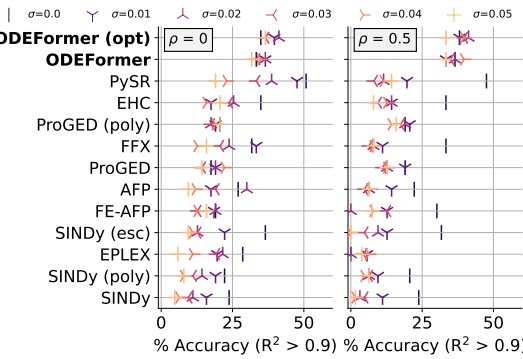

Figure 5: **Generalization on ODEBench.** We consider the same setting as in Figure 4.

## 6 CONCLUSION

We foresee real-world applications of ODEFormer across the sciences, for hypothesis generation of dynamical laws underlying experimental observations. However, in the following we also highlight several limitations of the current method, opening up interesting directions for future work.

First, we only considered first order ODEs. While any higher-order ODE can be written as a system of first order ODEs, this does not immediately allow ODEFormer to make predictions based only on a solution trajectory, since we would still need to approximate time derivatives up to the order of the ODE. While possible in principle via finite differencing schemes, we would suffer similar drawbacks as other methods that rely on finite differences from noisy, irregularly sampled data.

Second, ODEFormer only works when all variables are observed. In real-world settings, some relevant variables may be unknown or unobservable. For example, inferring chemical kinematics may be challenged by the difficulty in measuring the concentration of reaction intermediates (Burés & Larrosa, 2023). We may circumvent this issue by randomly masking variables during training (replacing their observations by a dedicated token), to emulate unobserved variables. While this raises questions around identifiability and robustness, we plan to explore masking in future work. This technique could also handle higher-order ODEs by considering the derivatives as unobserved.

Third, based on the four representatives in ODEBench, ODEFormer (as well as all other benchmarked models) struggles with chaotic systems, which have been argued to provide good assessments for data-driven modeling and forecasting (Gilpin, 2021). For chaotic systems, understanding properties of the attractor is often more desirable: nearby trajectories diverge exponentially, rendering identification notoriously challenging. Dynamical law learning from short (transient) trajectories for chaotic systems remains an interesting direction for future work.

Lastly, all existing methods for dynamical SR, including ours, perform inference based on a single observed trajectory. In our opinion, one of the most promising directions for future work is to enable inference from multiple solution trajectories of the same ODE. Key benefits may include averaging out possible noise sources as well as improving identifiability, as we "explore the domain of $f$". However, initial experiments with various forms of logit aggregation in ODEFormer's decoder during inference did not yield convincing results. In future work, we plan to exploit a cross-attention encoder to combine the embeddings of the different trajectories to leverage the combined information, in the spirit of Liu et al. (2023).

In addition to developing solutions to these limitation another exciting direction for future work concerns the extension of the framework towards other classes of differential equations, e.g. partial or stochastic DEs. Although conceptually straightforwards, finding scalable data generation procedures for these classes may be challenging - yet would considerably expand the set of possible applications.

We conclude by emphasizing that the difficulty and potential ambiguity in dynamical law learning calls for caution during deployment. Methods like ODEFormer primarily serve as hypothesis generators, ultimately requiring further experimental verification, something that can be done well based on ODEs (rather than black-box models).

## 7 REPRODUCIBILITY STATEMENT

The reproducibility of our work is ensured through several means. First and foremost, all code, model weights, and created benchmark datasets will be made publicly available at https://github.com/sdascoli/odeformer together with notebooks to directly reproduce key results, as well as a pip-installable package for easy usage. We also describe in detail the data generation in Section 3, our architecture and model choices in Section 4, and additional details about training in Appendices C and D.

## ACKNOWLEDGEMENTS

We thank Luca Biggio, François Charton and Tommaso Bendinelli for insightful conversations. We acknowledge funding from the EPFL's AI4science program (SA), Helmholtz Association under the joint research school "Munich School for Data Science - MUDS" (SB), Swiss National Science Foundation grant (310030_212516) (AM), NCCR Catalysis (grant number 180544), and a National Centre of Competence in Research funded by the Swiss National Science Foundation (PS). The authors gratefully acknowledge the Gauss Centre for Supercomputing e.V. (www.gauss-centre.eu) for funding this project (application number 29063, project "dynadis") by providing computing time on the GCS Supercomputer JUWELS (Jülich Supercomputing Centre, 2021) at Jülich Supercomputing Centre (JSC).

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

## A ODEBENCH

ODEBench features a selection of ordinary differential equations primarily from Steven Strogatz's book "Nonlinear Dynamics and Chaos" with manually chosen parameter values and initial conditions (Strogatz, 2000). Some other famous known systems have been selected from other sources such as Wikipedia, which are included in the dictionary entries as well. We selected ODEs primarily based on whether they have actually been suggested as models for real-world phenomena as well as on whether they are 'iconic' ODEs in the sense that they are often used as examples in textbooks and/or have recognizable names. Whenever there were 'realistic' parameter values suggested, we chose those.

In this benchmark, we typically include only one set of parameter values per equation. Many of the ODEs in Strogatz' book are analyzed in terms of the different limit behavior for different parameter settings. For some systems that exhibit wildly different behavior for different parameter settings, we include multiple sets of parameter values as separate equations (e.g., the Lorenz system in chaotic and non-chaotic regimes). For each equation, we include two sets of manually chosen initial conditions.

There are 23 equations in one dimension, 28 equations in two dimensions, 10 equations in three dimensions, and 2 equations in four dimensions. This results in a total of 63 equations, 4 of which display chaotic behavior. We provide the analytical expressions and initial conditions in Tables 2 to 4, visualizations of a single trajectory for each ODE in Figure 6, and ODEFormer's predictions for each ODE in Figure 7.

| ID | System description | Equation | Parameters | Initial values |
|----|-------------------|----------|------------|----------------|
| 1 | RC-circuit (charging capacitor) | $\dfrac{c_0 - \frac{x_0}{c_1}}{c_2}$ | 0.7, 1.2, 2.31 | [10.0], [3.54] |
| 2 | Population growth (naive) | $c_0 x_0$ | 0.23 | [4.78], [0.87] |
| 3 | Population growth with carrying capacity | $c_0 x_0 \cdot \left(1 - \frac{x_0}{c_1}\right)$ | 0.79, 74.3 | [7.3], [21.0] |
| 4 | RC-circuit with non-linear resistor (charging capacitor) | $-0.5 + \dfrac{1}{e^{c_0 - \frac{x_0}{c_1}} + 1}$ | 0.5, 0.96 | [0.8], [0.02] |
| 5 | Velocity of a falling object with air resistance | $c_0 - c_1 x_0^2$ | 9.81, 0.0021175 | [0.5], [73.0] |
| 6 | Autocatalysis with one fixed abundant chemical | $c_0 x_0 - c_1 x_0^2$ | 2.1, 0.5 | [0.13], [2.24] |
| 7 | Gompertz law for tumor growth | $c_0 x_0 \log\left(c_1 x_0\right)$ | 0.032, 2.29 | [1.73], [9.5] |
| 8 | Logistic equation with Allee effect | $c_0 x_0 \left(-1 + \frac{x_0}{c_2}\right)\left(1 - \frac{x_0}{c_1}\right)$ | 0.14, 130.0, 4.4 | [6.123], [2.1] |
| 9 | Language death model for two languages | $c_0 \cdot (1 - x_0) - c_1 x_0$ | 0.32, 0.28 | [0.14], [0.55] |
| 10 | Refined language death model for two languages | $c_0 x_0^{c_1} \cdot (1 - x_0) - x_0 \cdot (1 - c_0)(1 - x_0)^{c_1}$ | 0.2, 1.2 | [0.83], [0.34] |
| 11 | Naive critical slowing down (statistical mechanics) | $-x_0^3$ | | [3.4], [1.6] |
| 12 | Photons in a laser (simple) | $c_0 x_0 - c_1 x_0^2$ | 1.8, 0.1107 | [11.0], [1.3] |
| 13 | Overdamped bead on a rotating hoop | $c_0 \left(c_1 \cos\left(x_0\right) - 1\right)\sin\left(x_0\right)$ | 0.0981, 9.7 | [3.1], [2.4] |
| 14 | Budworm outbreak model with predation | $c_0 x_0 \cdot \left(1 - \frac{x_0}{c_1}\right) - \frac{c_3 x_0^2}{c_2^2 + x_0^2}$ | 0.78, 81.0, 21.2, 0.9 | [2.76], [23.3] |
| 15 | Budworm outbreak with predation (dimensionless) | $c_0 x_0 \cdot \left(1 - \frac{x_0}{c_1}\right) - \frac{x_0^2}{x_0^2 + 1}$ | 0.4, 95.0 | [44.3], [4.5] |
| 16 | Landau equation (typical time scale tau = 1) | $c_0 x_0 - c_1 x_0^3 - c_2 x_0^5$ | 0.1, -0.04, 0.001 | [0.94], [1.65] |
| 17 | Logistic equation with harvesting/fishing | $c_0 x_0 \cdot \left(1 - \frac{x_0}{c_1}\right) - c_2$ | 0.4, 100.0, 0.3 | [14.3], [34.2] |
| 18 | Improved logistic equation with harvesting/fishing | $c_0 x_0 \cdot \left(1 - \frac{x_0}{c_1}\right) - \frac{c_2 x_0}{c_3 + x_0}$ | 0.4, 100.0, 0.24, 50.0 | [21.1], [44.1] |
| 19 | Improved logistic equation with harvesting/fishing (dimensionless) | $-\frac{c_0 x_0}{c_1 + x_0} + x_0 \cdot (1 - x_0)$ | 0.08, 0.8 | [0.13], [0.03] |
| 20 | Autocatalytic gene switching (dimensionless) | $c_0 - c_1 x_0 + \frac{x_0^2}{x_0^2 + 1}$ | 0.1, 0.55 | [0.002], [0.25] |
| 21 | Dimensionally reduced SIR infection model for dead people (dimensionless) | $c_0 - c_1 x_0 - e^{-x_0}$ | 1.2, 0.2 | [0.0], [0.8] |
| 22 | Hysteretic activation of a protein expression (positive feedback, basal promoter expression) | $c_0 + \frac{c_1 x_0^5}{c_2 + x_0^5} - c_3 x_0$ | 1.4, 0.4, 123.0, 0.89 | [3.1], [6.3] |
| 23 | Overdamped pendulum with constant driving torque/fireflies/Josephson junction (dimensionless) | $c_0 - \sin\left(x_0\right)$ | 0.21 | [-2.74], [1.65] |

Table 2: Scalar ODEs in ODEBench.

| ID | System description | Equation | Parameters | Initial values |
|---|---|---|---|---|
| 24 | Harmonic oscillator without damping | $\begin{cases} x_1 \\ -c_0 x_0 \end{cases}$ | 2.1 | [0.4, -0.03], [0.0, 0.2] |
| 25 | Harmonic oscillator with damping | $\begin{cases} x_1 \\ -c_0 x_0 - c_1 x_1 \end{cases}$ | 4.5, 0.43 | [0.12, 0.043], [0.0, -0.3] |
| 26 | Lotka-Volterra competition model (Strogatz version with sheeps and rabbits) | $\begin{cases} x_0 (c_0 - c_1 x_1 - x_0) \\ x_1 (c_2 - x_0 - x_1) \end{cases}$ | 3.0, 2.0, 2.0 | [5.0, 4.3], [2.3, 3.6] |
| 27 | Lotka-Volterra simple (as on Wikipedia) | $\begin{cases} x_0 (c_0 - c_1 x_1) \\ -x_1 (c_2 - c_3 x_0) \end{cases}$ | 1.84, 1.45, 3.0, 1.62 | [8.3, 3.4], [0.4, 0.65] |
| 28 | Pendulum without friction | $\begin{cases} x_1 \\ -c_0 \sin(x_0) \end{cases}$ | 0.9 | [-1.9, 0.0], [0.3, 0.8] |
| 29 | Dipole fixed point | $\begin{cases} c_0 x_0 x_1 \\ -x_0^2 + x_1^2 \end{cases}$ | 0.65 | [3.2, 1.4], [1.3, 0.2] |
| 30 | RNA molecules catalyzing each others replication | $\begin{cases} x_0 (-c_0 x_0 x_1 + x_1) \\ x_1 (-c_0 x_0 x_1 + x_0) \end{cases}$ | 1.61 | [0.3, 0.04], [0.1, 0.21] |
| 31 | SIR infection model only for healthy and sick | $\begin{cases} -c_0 x_0 x_1 \\ c_0 x_0 x_1 - c_1 x_1 \end{cases}$ | 0.4, 0.314 | [7.2, 0.98], [20.0, 12.4] |
| 32 | Damped double well oscillator | $\begin{cases} x_1 \\ -c_0 x_1 - x_0^3 + x_0 \end{cases}$ | 0.18 | [-1.8, -1.8], [5.8, 0.0] |
| 33 | Glider (dimensionless) | $\begin{cases} -c_0 x_0^2 - \sin(x_1) \\ x_0 - \frac{\cos(x_1)}{x_0} \end{cases}$ | 0.08 | [5.0, 0.7], [9.81, -0.8] |
| 34 | Frictionless bead on a rotating hoop (dimensionless) | $\begin{cases} x_1 \\ (-c_0 + \cos(x_0)) \sin(x_0) \end{cases}$ | 0.93 | [2.1, 0.0], [-1.2, -0.2] |
| 35 | Rotational dynamics of an object in a shear flow | $\begin{cases} \cos(x_0) \cot(x_1) \\ \left(c_0 \sin^2(x_1) + \cos^2(x_1)\right) \sin(x_0) \end{cases}$ | 4.2 | [1.13, -0.3], [2.4, 1.7] |
| 36 | Pendulum with non-linear damping, no driving (dimensionless) | $\begin{cases} x_1 \\ -c_0 x_1 \cos(x_0) - x_1 - \sin(x_0) \end{cases}$ | 0.07 | [0.45, 0.9], [1.34, -0.8] |
| 37 | Van der Pol oscillator (standard form) | $\begin{cases} x_1 \\ -c_0 x_1 \left(x_0^2 - 1\right) - x_0 \end{cases}$ | 0.43 | [2.2, 0.0], [0.1, 3.2] |
| 38 | Van der Pol oscillator (simplified form from Strogatz) | $\begin{cases} c_0 \left(-\frac{x_0^3}{3} + x_0 + x_1\right) \\ -\frac{x_0}{c_0} \end{cases}$ | 3.37 | [0.7, 0.0], [-1.1, -0.7] |
| 39 | Glycolytic oscillator, e.g., ADP and F6P in yeast (dimensionless) | $\begin{cases} c_0 x_1 + x_0^2 x_1 - x_0 \\ -c_0 x_0 + c_1 - x_0^2 x_1 \end{cases}$ | 2.4, 0.07 | [0.4, 0.31], [0.2, -0.7] |
| 40 | Duffing equation (weakly non-linear oscillation) | $\begin{cases} x_1 \\ c_0 x_1 \cdot \left(1 - x_0^2\right) - x_0 \end{cases}$ | 0.886 | [0.63, -0.03], [0.2, 0.2] |
| 41 | Cell cycle model by Tyson for interaction between protein cdc2 and cyclin (dimensionless) | $\begin{cases} c_0 \left(c_1 + x_0^2\right)(-x_0 + x_1) - x_0 \\ c_2 - x_0 \end{cases}$ | 15.3, 0.001, 0.3 | [0.8, 0.3], [0.02, 1.2] |
| 42 | Reduced model for chlorine dioxide-iodine-malonic acid rection (dimensionless) | $\begin{cases} c_0 - \frac{c_1 x_0 x_1}{x_0^2 + 1} - x_0 \\ c_2 x_0 \left(-\frac{x_1}{x_0^2 + 1} + 1\right) \end{cases}$ | 8.9, 4.0, 1.4 | [0.2, 0.35], [3.0, 7.8] |
| 43 | Driven pendulum with linear damping / Josephson junction (dimensionless) | $\begin{cases} x_1 \\ c_0 - c_1 x_1 - \sin(x_0) \end{cases}$ | 1.67, 0.64 | [1.47, -0.2], [-1.9, 0.03] |
| 44 | Driven pendulum with quadratic damping (dimensionless) | $\begin{cases} x_1 \\ c_0 - c_1 x_1 \lvert x_1 \rvert - \sin(x_0) \end{cases}$ | 1.67, 0.64 | [1.47, -0.2], [-1.9, 0.03] |
| 45 | Isothermal autocatalytic reaction model by Gray and Scott 1985 (dimensionless) | $\begin{cases} c_0 \cdot (1 - x_0) - x_0 x_1^2 \\ -c_1 x_1 + x_0 x_1^2 \end{cases}$ | 0.5, 0.02 | [1.4, 0.2], [0.32, 0.64] |
| 46 | Interacting bar magnets | $\begin{cases} c_0 \sin(x_0 - x_1) - \sin(x_0) \\ -c_0 \sin(x_0 - x_1) - \sin(x_1) \end{cases}$ | 0.33 | [0.54, -0.1], [0.43, 1.21] |
| 47 | Binocular rivalry model (no oscillations) | $\begin{cases} -x_0 + \frac{1}{e^{c_0 x_1 - c_1} + 1} \\ -x_1 + \frac{1}{e^{c_0 x_0 - c_1} + 1} \end{cases}$ | 4.89, 1.4 | [0.65, 0.59], [3.2, 10.3] |
| 48 | Bacterial respiration model for nutrients and oxygen levels | $\begin{cases} c_0 - \frac{x_0 x_1}{c_1 x_0^2 + 1} - x_0 \\ c_2 - \frac{x_0 x_1}{c_1 x_0^2 + 1} \end{cases}$ | 18.3, 0.48, 11.23 | [0.1, 30.4], [13.2, 5.21] |
| 49 | Brusselator: hypothetical chemical oscillation model (dimensionless) | $\begin{cases} c_1 x_0^2 x_1 - x_0 (c_0 + 1) + 1 \\ c_0 x_0 - c_1 x_0^2 x_1 \end{cases}$ | 3.03, 3.1 | [0.7, -1.4], [2.1, 1.3] |
| 50 | Chemical oscillator model by Schnackenberg 1979 (dimensionless) | $\begin{cases} c_0 + x_0^2 x_1 - x_0 \\ c_1 - x_0^2 x_1 \end{cases}$ | 0.24, 1.43 | [0.14, 0.6], [1.5, 0.9] |
| 51 | Oscillator death model by Ermentrout and Kopell 1990 | $\begin{cases} c_0 + \sin(x_1) \cos(x_0) \\ c_1 + \sin(x_1) \cos(x_0) \end{cases}$ | 1.432, 0.972 | [2.2, 0.67], [0.03, -0.12] |

Table 3: 2 dimensional ODEs in ODEBench.

| ID | System description | Equation | Parameters | Initial values |
|---|---|---|---|---|
| 52 | Maxwell-Bloch equations (laser dynamics) | $c_0\left(-x_0 + x_1\right)$ 
 $c_1\left(x_0 x_2 - x_1\right)$ 
 $c_2\left(-c_3 x_0 x_1 + c_3 - x_2 + 1\right)$ | 0.1, 0.21, 0.34, 3.1 | [1.3, 1.1, 0.89], [0.89, 1.3, 1.1] |
| 53 | Model for apoptosis (cell death) | $c_0 - c_4 x_0 - \frac{c_5 x_0 x_1}{c_9 + x_0}$ 
 $c_1 x_2\left(c_8 + x_1\right) - \frac{c_2 x_1}{c_6 + x_1} - \frac{c_3 x_0 x_1}{c_7 + x_1}$ 
 $-c_1 x_2\left(c_8 + x_1\right) + \frac{c_2 x_1}{c_6 + x_1} + \frac{c_3 x_0 x_1}{c_7 + x_1}$ | 0.1, 0.6, 0.2, 7.95, 0.05, 0.4, 0.1, 2.0, 0.1, 0.1 | [0.005, 0.26, 2.15], [0.248, 0.0973, 0.0027] |
| 54 | Lorenz equations in well-behaved periodic regime | $c_0\left(-x_0 + x_1\right)$ 
 $c_1 x_0 - x_0 x_2 - x_1$ 
 $-c_2 x_2 + x_0 x_1$ | 5.1, 12.0, 1.67 | [2.3, 8.1, 12.4], [10.0, 20.0, 30.0] |
| 55 | Lorenz equations in complex periodic regime | $c_0\left(-x_0 + x_1\right)$ 
 $c_1 x_0 - x_0 x_2 - x_1$ 
 $-c_2 x_2 + x_0 x_1$ | 10.0, 99.96, 8/3 | [2.3, 8.1, 12.4], [10.0, 20.0, 30.0] |
| 56 | Lorenz equations standard parameters (chaotic) | $c_0\left(-x_0 + x_1\right)$ 
 $c_1 x_0 - x_0 x_2 - x_1$ 
 $-c_2 x_2 + x_0 x_1$ | 10.0, 28.0, 8/3 | [2.3, 8.1, 12.4], [10.0, 20.0, 30.0] |
| 57 | Rössler attractor (stable fixed point) | $c_3\left(-x_1 - x_2\right)$ 
 $c_3\left(c_0 x_1 + x_0\right)$ 
 $c_3\left(c_1 + x_2\left(-c_2 + x_0\right)\right)$ | -0.2, 0.2, 5.7, 5.0 | [2.3, 1.1, 0.8], [-0.1, 4.1, -2.1] |
| 58 | Rössler attractor (periodic) | $c_3\left(-x_1 - x_2\right)$ 
 $c_3\left(c_0 x_1 + x_0\right)$ 
 $c_3\left(c_1 + x_2\left(-c_2 + x_0\right)\right)$ | 0.1, 0.2, 5.7, 5.0 | [2.3, 1.1, 0.8], [-0.1, 4.1, -2.1] |
| 59 | Rössler attractor (chaotic) | $c_3\left(-x_1 - x_2\right)$ 
 $c_3\left(c_0 x_1 + x_0\right)$ 
 $c_3\left(c_1 + x_2\left(-c_2 + x_0\right)\right)$ | 0.2, 0.2, 5.7, 5.0 | [2.3, 1.1, 0.8], [-0.1, 4.1, -2.1] |
| 60 | Aizawa attractor (chaotic) | $-c_3 x_1 + x_0\left(-c_1 + x_2\right)$ 
 $c_3 x_0 + x_1\left(-c_1 + x_2\right)$ 
 $c_0 x_2 + c_2 + c_5 x_0^3 x_2 - 1/3 x_2^3 - \left(x_0^2 + x_1^2\right)\left(c_4 x_2 + 1\right)$ | 0.95, 0.7, 0.65, 3.5, 0.25, 0.1 | [0.1, 0.05, 0.05], [-0.3, 0.2, 0.1] |
| 61 | Chen-Lee attractor; system for gyro motion with feedback control of rigid body (chaotic) | $c_0 x_0 - x_1 x_2$ 
 $c_1 x_1 + x_0 x_2$ 
 $c_2 x_2 + \frac{x_0 x_1}{c_3}$ | 5.0, -10.0, -3.8, 3.0 | [15, -15, -15], [8, 14, -10] |
| 62 | Binocular rivalry model with adaptation (oscillations) | $-x_0 + \frac{1}{e^{c_0 x_2 + c_1 x_1 - c_2} + 1}$ 
 $c_3\left(x_0 - x_1\right)$ 
 $-x_2 + \frac{1}{e^{c_0 x_0 + c_1 x_3 - c_2} + 1}$ 
 $c_3\left(x_2 - x_3\right)$ | 0.89, 0.4, 1.4, 1.0 | [2.25, -0.5, -1.13, 0.4], [0.342, -0.431, -0.86, 0.041] |
| 63 | SEIR infection model (proportions) | $-c_1 x_0 x_2$ 
 $-c_0 x_1 + c_1 x_0 x_2$ 
 $c_0 x_1 - c_2 x_2$ 
 $c_2 x_2$ | 0.47, 0.28, 0.3 | [0.6, 0.3, 0.09, 0.01], [0.4, 0.3, 0.25, 0.05] |

Table 4: 3 and 4 dimensional ODEs in ODEBench.

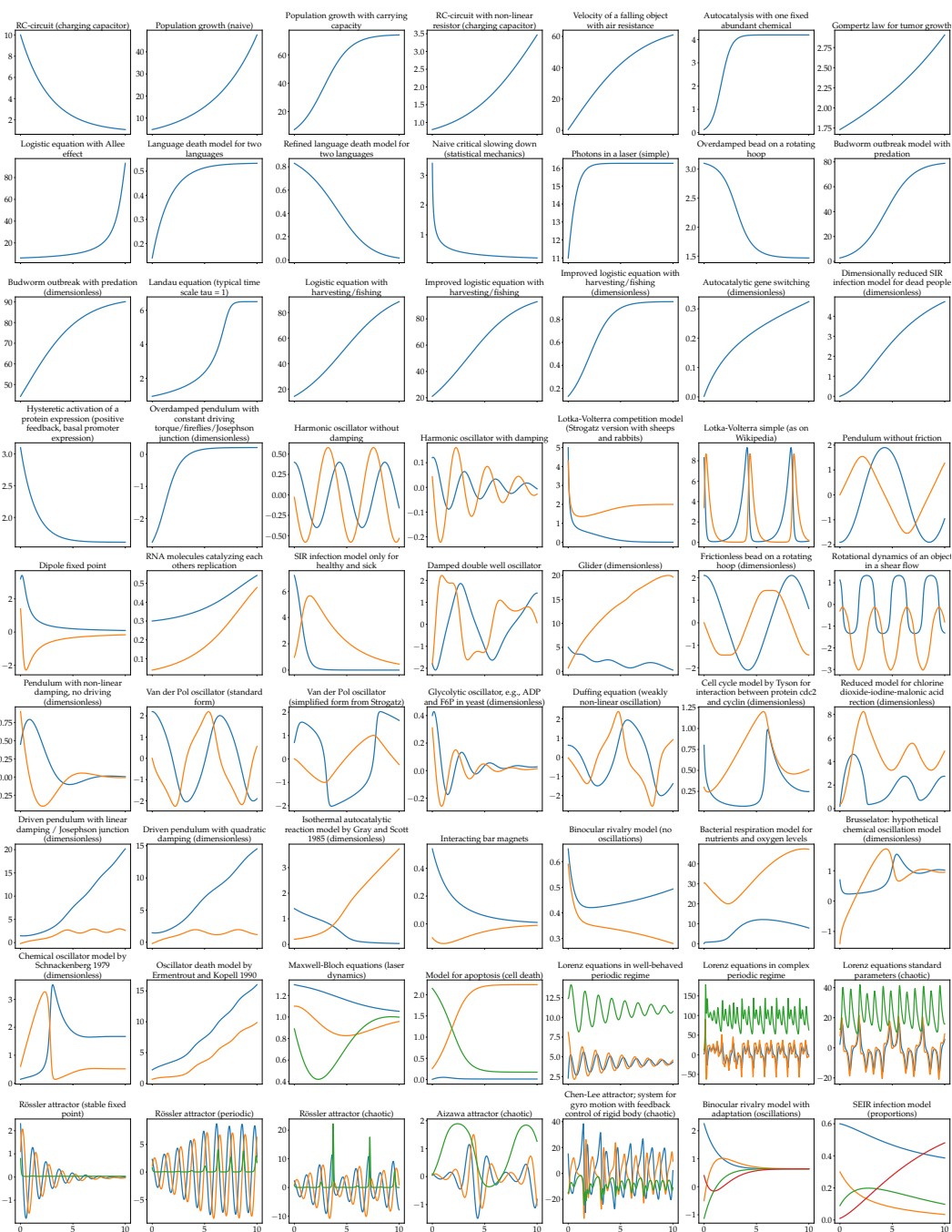

Figure 6: Solution trajectories of all equations in ODEBench for one of the initial conditions.

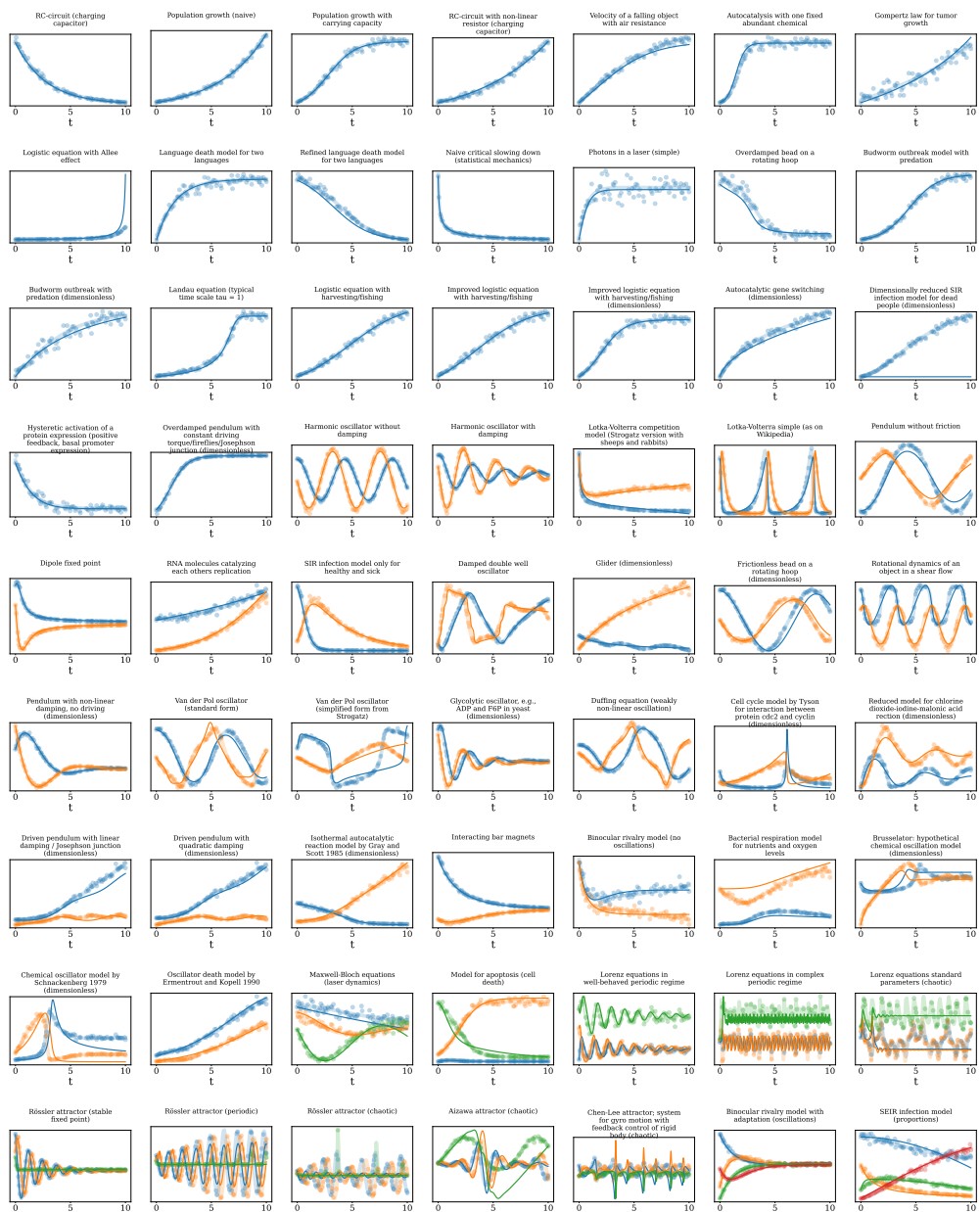

Figure 7: Predictions of ODEFormer for all equations in ODEBench for the first set of initial conditions.

## B   IDENTIFIABILITY

Traditionally, when inferring dynamical laws as ODEs from observations, one assumed a parametric form of the ODE to be known. In addition, typical assumptions included that the state of the system may not be fully observed, and that one may not be interested in identifying the system fully (all parameters), but that there are only certain (combinations of) parameters that need to be identified. For example, in a typical epidemiological model of disease spread, e.g., modelling the fractions of susceptible, infected, and recovered people (SIR model) in a population, one may only be able to measure the number of infected people, but may also only be interested in estimating the reproduction rate (a ratio of two parameters in the full equations). Hence a typical identifiability question would start from a known system, a known observation function, and a specific target quantity. Researchers have then developed various general methods and procedures to decide whether such a query is solvable, i.e., whether the target is identified from the observations within the assumptions (Åström & Eykhoff, 1971; Miao et al., 2011; Villaverde et al., 2016; Hamelin et al., 2020).

The study of identifiability from observations within larger non-parametric function classes from full state observations has only been taken up more recently. For example, linear autonomous systems as well as autonomous systems that are linear in parameters are fairly well understood (Stanhope et al., 2014; Duan et al., 2020) with Qiu et al. (2022) recently essentially providing closure to this question. A broad summary of these findings is that linear (in parameters) autonomous systems are almost always (almost surely) identifiable from a single trajectory for many reasonable measures (probability distributions) over parameters, i.e., over ODE systems.

When it comes to non-parametric classes such as analytic, algebraic, continuous, or smooth functions, Scholl et al. (2023) have recently presented the first detailed analysis and results for a broad class of scalar PDEs. These include ODEs as special cases, however their results only apply to scalar ODEs. Even though they consider ODEs beyond first order, this still does not include multivariate ODE systems. In the scalar case, identification from a single solution trajectory is possible for analytic functions $f$, but essentially impossible for continuous (or smooth) $f$ (Scholl et al., 2023).[6]

The function class considered by ODEFormer, i.e., the distribution implied by our generator, predominantly contains real analytic functions, but not exclusively since $x \mapsto 1/x$ is analytic only on $\mathbb{R} \setminus \{0\}$. Hence, it is not clear in which category we fall regarding identifiability.

Crucially, all these theoretical results assume the entire continuous, non-noisy solution trajectory to be known. Little is known for discrete and noisy observations, where identifiability likely turns from a yes/no question into one of probabilistic claims given a prior over functions and the assumed noise model. Hence, current theoretical results do not conclude whether non-linear ODEs can practically be inferred from data.

## C   TRAINING DETAILS FOR ODEFORMER

We optimize the cross-entropy loss between the predicted sequence of mathematical symbols and the target sequence. During training, (the right-hand side of) ODEs are represented in prefix format instead of the more familiar infix format in order to relieve the model from having to predict correct opening and closing parenthesis (as an example: "$3 \cdot (x_0 + x_1)$" corresponds to "$\cdot, 3, +, x_0, x_1$" in prefix notation). We note that different target sequences may be mathematically equivalent but do not attempt to resolve this ambiguity on the training data set as it does not hamper model optimization in practice.

We use the Adam optimizer (with default parameters suggested by Kingma & Ba (2015)), with a learning rate warming up from $10^{-7}$ to $2 \times 10^{-4}$ across the initial 10,000 steps and a subsequent decaying governed by a cosine schedule for the next 300,000 steps. The annealing cycle then restarts with a damping factor of 3/2 as per Kingma & Ba (2015), resulting in approximately 800,000 optimization steps. We do not use any regularization such as weight decay or dropout. To efficiently manage the greatly varying input sequence lengths, we group examples of similar lengths in batches, with the constraint that each batch contains 10,000 tokens. Our model is trained on a set of about

---

[6]The intuition here is that linear, polynomial, or more broadly even real analytic scalar univariate functions can be uniquely extrapolated to all of $\mathbb{R}$ when they are known on any open interval. On the other hand, there are infinitely many ways to extrapolate continuous or smooth functions beyond any interval.

50M examples pre-generated with 80 CPU cores. When run on a single NVIDIA A100 GPU with 80GB memory and 8 CPU cores, ODEFormer's training process takes roughly three days.

To enforce model robustness to signal corruptions we contaminate training trajectories $\mathbf{x}(\mathbf{t_i})$ with noise according to $x_j(t_i) \mapsto (1 + \xi)x_j(t_i)$ for $j \in [D], i \in [N]$ and $\xi \sim \mathcal{N}(0, \sigma)$.

An overview of the model architecture can be found in Table 6.

## D    OPTIONAL CONSTANT OPTIMIZATION IN ODEFORMER (OPT)

In contrast to all baseline models, ODEFormer is a pretrained model and predicted ODEs are not explicitly fit to the data observed at inference time. However, similar to Kamienny et al. (2022) we can post-hoc optimize the parameters of a predicted ODE to improve the data fit. Although parameter estimation for dynamical system is known to be a challenging inference problem, we use the Broyden-Fletcher-Goldfarb-Shanno algorithm (BFGS) (Nocedal & Wright, 2006) as implemented in `scipy.optimize.minimize` (Virtanen et al., 2020) and thus opt for a comparatively simple local, gradient-based method in the hope that the parameter values predicted by ODEFormer only need slight refinement. The optimizer solves the following problem

$$\arg\min_{\boldsymbol{\theta}} loss(\{x(t_0), \dots, x(t_n)\}, \texttt{solve\_ivp}(\hat{f}(x; \boldsymbol{\theta}), x_0 = x(t_0), t = \{t_0, \dots, t_n\}))$$

where $\{p_1, \dots, p_k\}$ denotes the set of parameters of the ODE $\hat{f}$ that was predicted by ODEFormer, and where $(x(t_0) \dots, x(t_n))$ represents the (potentially noisy) observations. We use the negative variance-weighted $R^2$ score as optimization loss.

## E    EVALUATION OF BASELINE MODELS

**Hyperparameter optimization.**    All baseline models are fitted to each trajectory separately and each fit involves a separate hyperparameter optimization. Hyperparameters that are searched over are listed in Table 5, all other hyperparameters are set to their respective default values. For each combination of hyperparameters, the model is fitted on the first 70% and scored on the remaining 30% of a trajectory. To reduce runtime, we parallelize optimization according to `GridSearchCV` from `scikit-learn` (Pedregosa et al., 2011) and set the number of parallel jobs to $\min($# combinations, # cpu cores (= 48)). After selecting the combination with highest $R^2$ score, the final model is fitted on the full trajectory.

**Finite difference approximations.**    Except for ProGED, all baseline models require approximations of temporal derivatives of all state variables of an ODE system as regression targets. To estimate temporal derivatives we use the central finite difference algorithm as implemented by `FiniteDifference` in the `pysindy` software package (de Silva et al., 2020) and include the approximation order in the hyperparameter search. For a fair comparison on noisy trajectories we extend the hyperparameter search to also include optional smoothing of trajectories with a Savitzky-Savgol filter with a window length of 15 as implemented by `SmoothedFiniteDifference` (de Silva et al., 2020).

**Vector-valued functions.**    Some of the baseline implementations (AFP, FE-AFP, EPLEX, EHC, FFX) do not readily support vector-valued functions ($f : \mathbb{R}^D \to \mathbb{R}^D$) but only scalar-valued functions ($f : \mathbb{R}^D \to \mathbb{R}$). To evaluate these baselines on systems of ODEs, we run them separately for each component $f_i : \mathbb{R}^D \to \mathbb{R}$ of the system and combine the predictions for all components $i \in \{1, \dots, D\}$ via the Cartesian product $\{f_1^1, \dots, f_1^{K_1}\} \times \dots \times \{f_D^1, \dots, f_D^{K_D}\}$ where $K_i$ represents the number of predictions, e.g., the length of the Pareto front, obtained for component $i$.

**Candidate selection.**    In symbolic regression, one typically faces a trade-off between accuracy (how well the function/trajectory is recovered) and complexity of the proposed expression. There are different strategies in the literature to select a single, final equation from the accuracy-complexity Pareto front, which may bias comparisons across methods along one or the other dimension. For a fair comparison, we evaluate all equations of a model's Pareto front and pick the final equation based on $R^2$ score between (potentially noisy) observed trajectory and the integrated solution of the predicted ODE.

Table 5: Hyperparameter names and values for optimization of baseline models. For FFX and PySR, we optimize over finite difference order and smoother window length but no additional hyper parameters.

| Model | Hyperparameter | Values |
|---|---|---|
| **All models** | finite difference order
smoother window length | 2, 3, 4
None, 15 |
| **AFP** | population size
generations
operators | 100, 500, 1000
2500, 500, 250
[n, v, +, -, *, /, exp, log, 2, 3, sqrt],
[n, v, +, -, *, /, exp, log, 2, 3, sqrt, sin, cos] |
| **EHC** | population size
generations
operators | 100, 500, 1000
1000, 200, 100
[n, v, +, -, *, /, exp, log, 2, 3, sqrt],
[n, v, +, -, *, /, exp, log, 2, 3, sqrt, sin, cos] |
| **EPLEX** | population size
generations
operators | 100, 500, 1000
2500, 500, 250
[n, v, +, -, *, /, exp, log, 2, 3, sqrt],
[n, v, +, -, *, /, exp, log, 2, 3, sqrt, sin, cos] |
| **FE-AFP** | population size
generations
operators | 100, 500, 1000
2500, 500, 250
[n, v, +, -, *, /, exp, log, 2, 3, sqrt],
[n, v, +, -, *, /, exp, log, 2, 3, sqrt, sin, cos] |
| **ProGED** | grammar | universal, rational
simplerational, trigonometric
polynomial |
| **SINDy** | polynomial degree
basis functions

optimizer threshold
optimizer alpha
optimizer max iterations | 1, 2, 3, 4, 5, 6, 7, 8, 9, 10
[polynomials],
[polynomials, sin, cos, exp],
[polynomials, sin, cos, exp, log, sqrt, 1/x]
0.05, 0.1, 0.15
0.025, 0.05, 0.075
20, 100 |
| **SINDy (esc)** | polynomial degree
basis functions

optimizer threshold
optimizer alpha
optimizer max iterations | 1, 2, 3, 4, 5, 6, 7, 8, 9, 10
[polynomials],
[polynomials, sin, cos, exp]
0.05, 0.1, 0.15
0.025, 0.05, 0.075
20, 100 |
| **SINDy (poly)** | polynomial degree
optimizer threshold
optimizer alpha
optimizer max iterations | 1, 2, 3, 4, 5, 6, 7, 8, 9, 10
0.05, 0.1, 0.15
0.025, 0.05, 0.075
20, 100 |

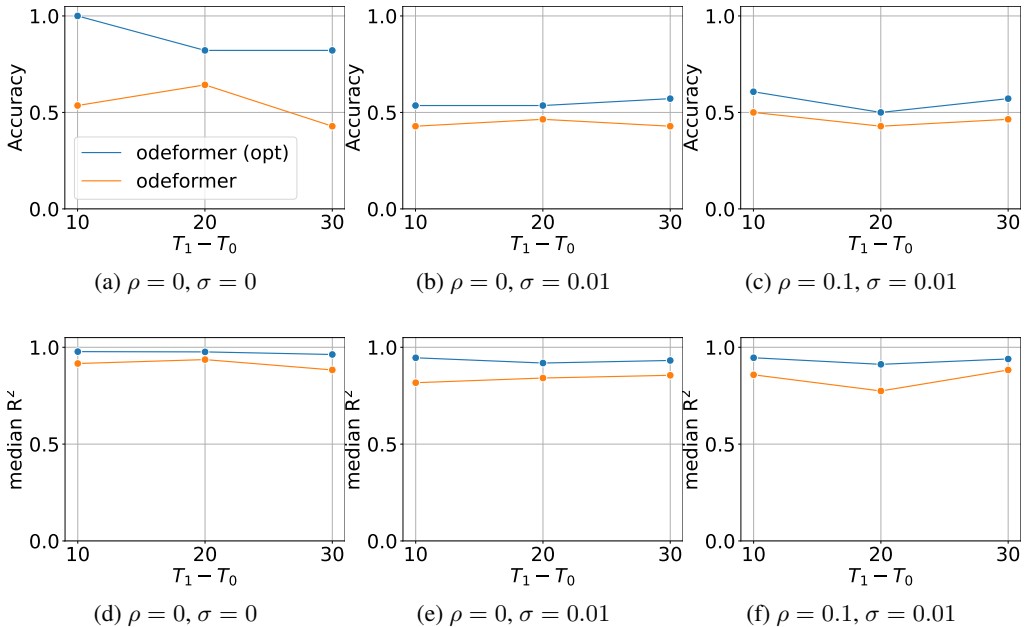

Figure 8: **Evaluation on Strogatz with missing chunks of data.** We compare ODEFormer with and without constant optimization (opt). Originally, trajectories in the Strogatz dataset comprise 100 time points. For this evaluation we drop all data in the interval $[T_0, T_1]$ and additionally drop a fraction of $\rho$ of the remaining time points uniformly at random. For $\sigma > 0$ the trajectories are additionally contaminated with noise. Plots in the upper row show results in terms of accuracy (fraction of samples with $R^2 > 0.9$) whereas the lower row shows the median $R^2$ score across the dataset.

# F  INTERPOLATION EXPERIMENTS FOR MISSING DATA CHUNKS

Whereas in the main text we considered irregular grids and corruptions by Gaussian noise, here we additionally evaluate ODEFormer in case of missing chunks of data. This scenario may for instance correspond to a faulty sensor which stops recording data for a duration of time, e.g. until it gets fixed or replaced.

To simulate this scenario we use the Strogatz dataset. Each trajectory in this dataset covers the time interval $[0, 10]$ and is originally comprised of 100 time points. Out of these 100 time points, we drop all time points within the interval $[T_0, T_1]$ for $[T_0, T_1] \in \{[45, 55], [40, 60], [35, 65]\}$, i.e., we assess model performance on three progressively more challenging missing data scenarios. Additionally, we corrupt the remaining data by randomly dropping observation and noise contamination as described in Section 5. We evaluate the same (trained) model as in Section 5. We remark that this model has not been optimized for this particular scenario as during training time points were dropped uniformly at random across the entire trajectory. Thus, intervals of entirely missing data may have been encountered in during training in exceptionally rare cases only.

Results under different signal corruptions are presented in Figure 8 while exemplary trajectories and corresponding predictions are displayed in Figure 9. Even under this challenging scenario, ODEFormer still performs reasonably well for the majority of samples as exemplified by both accuracy and median $R^2$ scores.

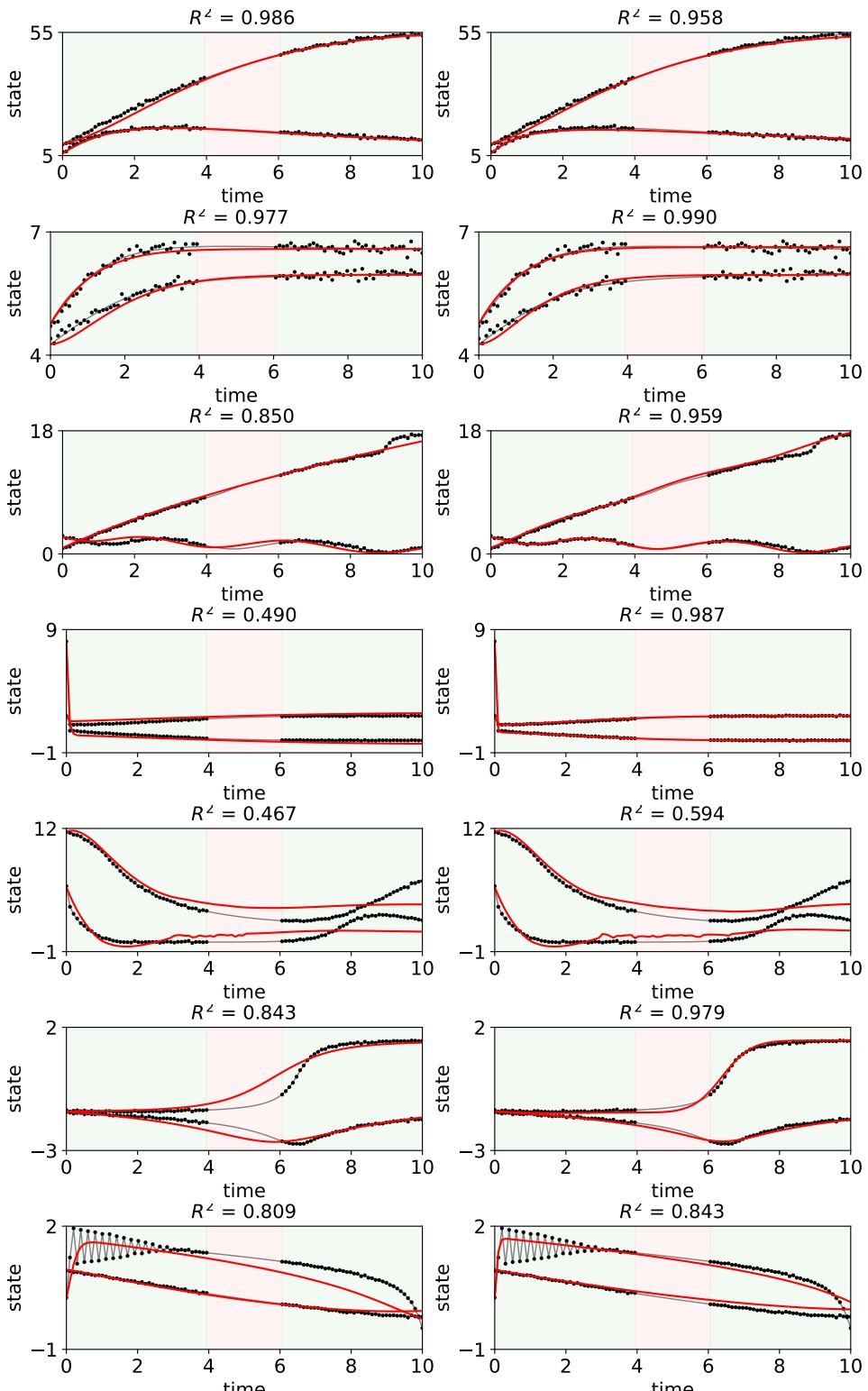

Figure 9: Predictions of ODEFormer (left column) and ODEFormer (opt) (right column) on noisy observations (black markers) when data is missing in a consecutive interval (red area). The thin gray line corresponds to the true (but unobserved) solution trajectory, the red line corresponds to the trajectory given by integrating the predicted ODE. Each row corresponds to one of the seven ODEs available in dataset Strogatz.

## G    EFFECT OF BEAM SIZE

In Figure 10, we study the impact of the beam size on reconstruction and generalization performance. While reconstruction improves with the beam size, generalization hardly changes. This highlights the importance of using both metrics: the two are not necessarily correlated, and the latter is a much better proxy of symbolic recovery than the former.

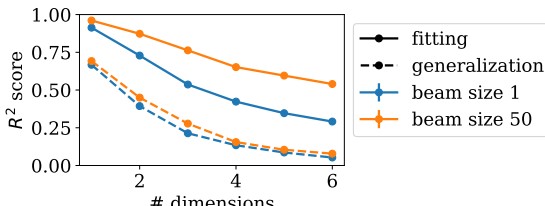

Figure 10: **Increasing the beam size improves reconstruction, but not generalization.** We plot the average reconstruction and generalization $R^2$-score on 10,000 noise-free, densely samples synthetic examples for various beam sizes and a temperature of 0.1.

## H    DETAILED COMPARISON WITH CLOSELY RELATED MODELS

### H.1    SIMILARITIES AND DIFFERENCES

As mentioned in Section 2, ODEFormer is conceptually similar to the models proposed by d'Ascoli et al. (2022) and Becker et al. (2023). Like ODEFormer, these models leverage large-scale training on synthetically generated samples to learn a transformer model that maps numerical input sequences to symbolic output sequences. While the output for ODEFormer and NSODE (the model presented by Becker et al. (2023)) correspond to (the right-hand side of) symbolic ordinary differential equations (ODEs), the model by d'Ascoli et al. (2022) predicts the discrete recurrence relation of the input sequence. A major difference between ODEFormer and these two related models is that ODEFormer works with multivariate inputs, i.e., it is able to handle systems of differential equations, whereas both other models only support univariate input sequences. For this reason neither NSODE nor the model by d'Ascoli et al. (2022) include an embedding module that needs to combine multivariate inputs as described in Section 4.

While the recurrent sequence prediction approach outlined by d'Ascoli et al. (2022) already delineates a distinct use-case, the divergence between ODEFormer and NSODE is especially noteworthy from a practical perspective. While multivariate systems of ODEs can capture rich dynamic behaviors (including e.g. limit cycles), univariate ODEs are somewhat limited in that they can only capture monotonically converging or monotonically diverging behavior. Moreover, the majority of intricate modeling scenarios typically involves multiple dynamically interacting variables, rendering a univariate model simply inadequate. Consequently, ODEFormer significantly broadens the practical toolset for ODE modelers in practice. We list the technical differences between all three models in Table 6.

Table 6: Model architecture of ODEFormer in comparison to closely related models.

|  |  | ODEFormer (ours) | Becker et al. (2023) | d'Ascoli et al. (2022) |
|---|---|---|---|---|
|  | Developed for | multivariate ODE systems | univariate ODEs | discrete univariate recurrence sequences |
| Encoder | # layers | 4 | 6 | 8 |
|  | # attn. heads | 16 | 16 | 8 |
|  | embedding dim. | 512 | 512 | 512 |
|  | forward dim. | 512 | 2048 | 512 |
|  | positional enc. | - | learned | - |
| Decoder | # layers | 16 | 6 | 8 |
|  | # attn. heads | 16 | 16 | 8 |
|  | embedding dim. | 512 | 512 | 512 |
|  | forward dim. | 512 | 2048 | 512 |
|  | beam size | 50 | 1536 | 10 |
|  | constants enc. | sign, mantissa, exp. | two-hot | sign, mantissa, exp. |

## H.2 Performance comparison

**Experimental setup.** As NSODE can not be applied to multivariate systems, we can not evaluate its performance on Strogatz and ODEBench. Instead, we evaluate ODEFormer on the datasets used in Becker et al. (2023), following their exact experimental conditions, which we briefly restate here for self-containment:

- The evaluation includes three datasets: **Textbook** contains 12 ODEs transcribed from physics textbooks, **Classic** contains 26 ODEs which correspond to classic (algebraic) equations in the symbolic regression literature which are re-interpreted as ODEs and lastly **Large** comprises 163 ODEs which are synthetically generated following the algorithm of Lample & Charton (2019). Although this dataset is generated following a similar procedure as described in Section 3, we expect the distribution of the generated data to be quite different from ODEFormer's training distribution as essentially all hyper-parameters (e.g. maximum. tree depth, operator probabilities, probability to sample a variable vs. a constant) differ. For details on the construction of the Large dataset, see section 3.1 in Becker et al. (2023).
- All equations are integrated in the interval $[0, 2]$ with 512 steps using `scipy.integrate.odeint` (Virtanen et al., 2020) to interface the LSODA (Hindmarsh & Laboratory, 1982) solver package. Numerical solution are subsequently systematically or randomly subsampled, resulting in an equidistant or irregular sampling grid. We remark that trajectories with 256 time points contain more time points than ODEFormer has seen for any trajectory during training (max. 200, see Section 3). Furthermore the numerical solver differs from the one used for generating the training data according to Section 3.
- Trajectories are corrupted with multiplicative noise drawn from $\mathcal{N}(1, \sigma)$.
- Evaluation always occurs on the original non-subsampled and noise-free solution trajectory.

Results for ODEFormer are obtained with a beam size of 50 and without additional constant optimization. Performance is evaluated in terms of median $R^2$ (which is the main metric used in Becker et al. (2023)) as well as accuracy as defined in Section 5, that is, the fraction of predictions for which $R^2 > 0.9$.

**Results.** The results in Tables 7 to 9 show that across all three datasets and all experimental conditions, ODEFormer performs quite similar to NSODE. In particular on **Textbook** and **Large** both models achieve strong results with negligible differences in median $R^2$ yet slightly superior performance of NSODE in terms of accuracy. This trend is also observable on dataset **Large** where differences in median $R^2$ are again small in most cases yet differences in accuracy are more pronounced in favor of NSODE. This result is expected given that equations in **Large** are generated from the same distribution that NSODE is trained on while the training distribution for ODEFormer differs. Overall, we conclude that ODEFormer generalizes well to different test time distributions.

Table 7: Results on **Textbook**

| model | #points | sampling | $\sigma$ (noise) | median $R^2$ | accuracy |
|---|---|---|---|---|---|
| ODEFormer | 128 | equidistant | 0 | 0.99 | 1 |
| NSODE | 128 | equidistant | 0 | 0.99 | 1 |
| ODEFormer | 128 | irregular | 0.01 | 0.99 | 0.91 |
| NSODE | 128 | irregular | 0.01 | 0.999 | 1 |
| ODEFormer | 128 | irregular | 0.02 | 0.994 | 0.818 |
| NSODE | 128 | irregular | 0.02 | 0.988 | 0.91 |
| ODEFormer | 256 | equidistant | 0 | 0.999 | 1 |
| NSODE | 256 | equidistant | 0 | 0.997 | 1 |
| ODEFormer | 256 | irregular | 0.01 | 0.994 | 1 |
| NSODE | 256 | irregular | 0.01 | 0.999 | 1 |
| ODEFormer | 256 | irregular | 0.02 | 0.993 | 1 |
| NSODE | 256 | irregular | 0.02 | 0.987 | 1 |

Table 8: Results on **Classic**

| model | #points | sampling | $\sigma$ (noise) | median $R^2$ | accuracy |
|---|---|---|---|---|---|
| ODEFormer | 128 | equidistant | 0 | 0.999 | 0.92 |
| NSODE | 128 | equidistant | 0 | 0.999 | 1 |
| ODEFormer | 128 | irregular | 0.01 | 0.998 | 0.92 |
| NSODE | 128 | irregular | 0.01 | 0.999 | 1 |
| ODEFormer | 128 | irregular | 0.02 | 0.998 | 0.84 |
| NSODE | 128 | irregular | 0.02 | 0.997 | 0.88 |
| ODEFormer | 256 | equidistant | 0 | 0.999 | 0.92 |
| NSODE | 256 | equidistant | 0 | 0.999 | 1 |
| ODEFormer | 256 | irregular | 0.01 | 0.998 | 0.88 |
| NSODE | 256 | irregular | 0.01 | 0.999 | 1 |
| ODEFormer | 256 | irregular | 0.02 | 0.998 | 0.84 |
| NSODE | 256 | irregular | 0.02 | 0.993 | 0.92 |

Table 9: Results on **Large**

| model | #points | sampling | $\sigma$ (noise) | median $R^2$ | accuracy |
|---|---|---|---|---|---|
| ODEFormer | 128 | equidistant | 0 | 0.978 | 0.652 |
| NSODE | 128 | equidistant | 0 | 0.998 | 0.789 |
| ODEFormer | 128 | irregular | 0.01 | 0.961 | 0.571 |
| NSODE | 128 | irregular | 0.01 | 0.997 | 0.776 |
| ODEFormer | 128 | irregular | 0.02 | 0.933 | 0.565 |
| NSODE | 128 | irregular | 0.02 | 0.982 | 0.671 |
| ODEFormer | 256 | equidistant | 0 | 0.996 | 0.64 |
| NSODE | 256 | equidistant | 0 | 0.998 | 0.801 |
| ODEFormer | 256 | irregular | 0.01 | 0.99 | 0.627 |
| NSODE | 256 | irregular | 0.01 | 0.999 | 0.82 |
| ODEFormer | 256 | irregular | 0.02 | 0.923 | 0.522 |
| NSODE | 256 | irregular | 0.02 | 0.991 | 0.745 |

# I    ADDITIONAL RESULTS ON BENCHMARKS

In Figures 11 to 16, we plot histograms to better visualize how each model performs across different noise conditions, datasets, and evaluation tasks.

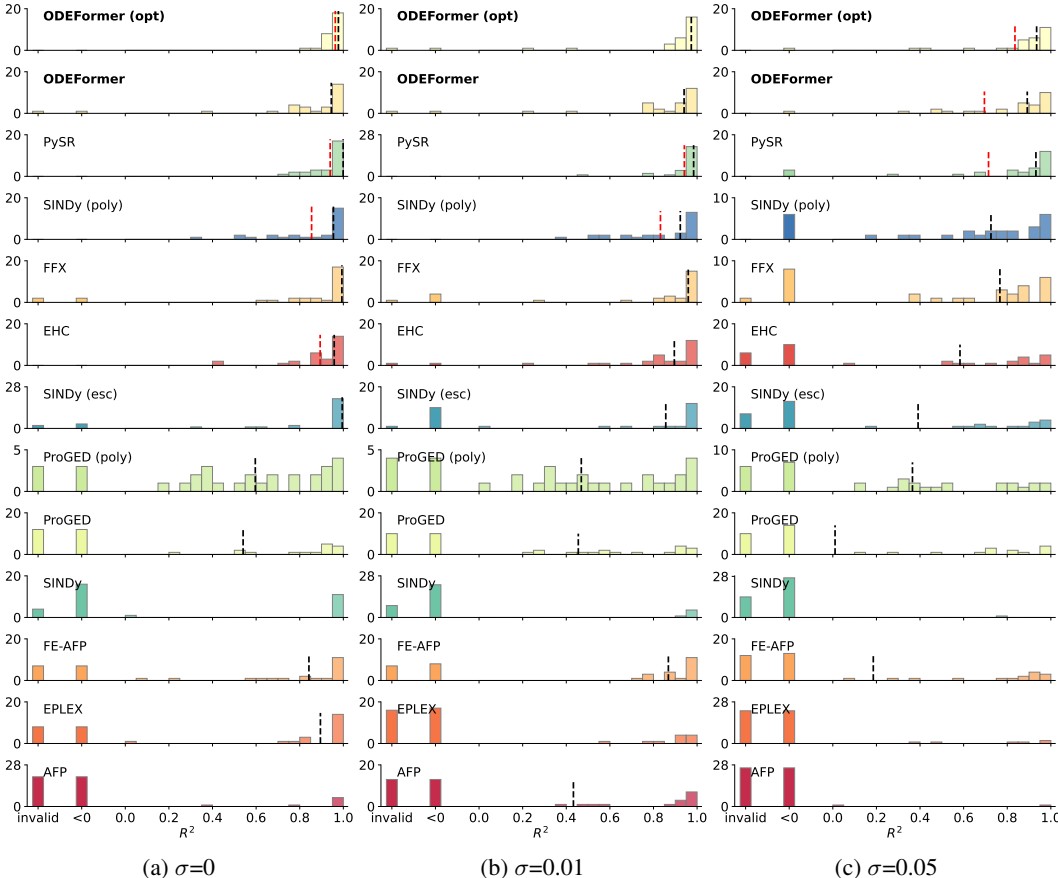

(a) $\sigma$=0                    (b) $\sigma$=0.01                    (c) $\sigma$=0.05

Figure 11: Histogram of per equation $R^2$ scores for the **reconstruction** task on **Strogatz**. Subfigures correspond to different noise levels. The y-axis represents counts and is scaled per model for better visibility of the distribution of scores. The x-axis annotations "invalid" and "<0" respectively denote the number of invalid predictions as well as the number of predictions that yielded an $R^2$ score below 0. The red dashed line corresponds to the mean $R^2$ score across equations, the black dashed line corresponds to the median $R^2$ score.

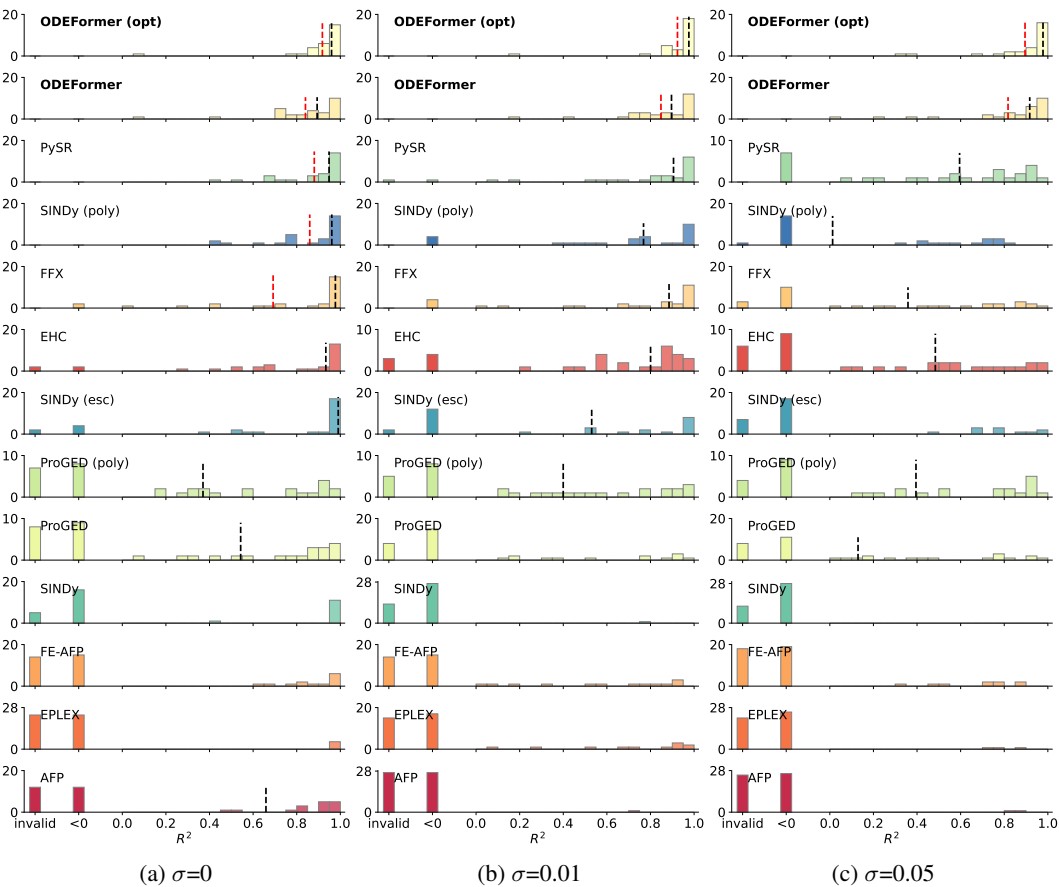

(a) $\sigma=0$        (b) $\sigma=0.01$        (c) $\sigma=0.05$

Figure 12: Histogram of per equation $R^2$ scores for the **reconstruction** task on **Strogatz** where **50%** of the trajectory are dropped uniformly at random ($\rho = 0.5$). Subfigures correspond to different noise levels. The y-axis represents counts and is scaled per model for better visibility of the distribution of scores. The x-axis annotations "invalid" and "<0" respectively denote the number of invalid predictions as well as the number of predictions that yielded an $R^2$ score below 0. The red dashed line corresponds to the mean $R^2$ score across equations, the black dashed line corresponds to the median $R^2$ score.

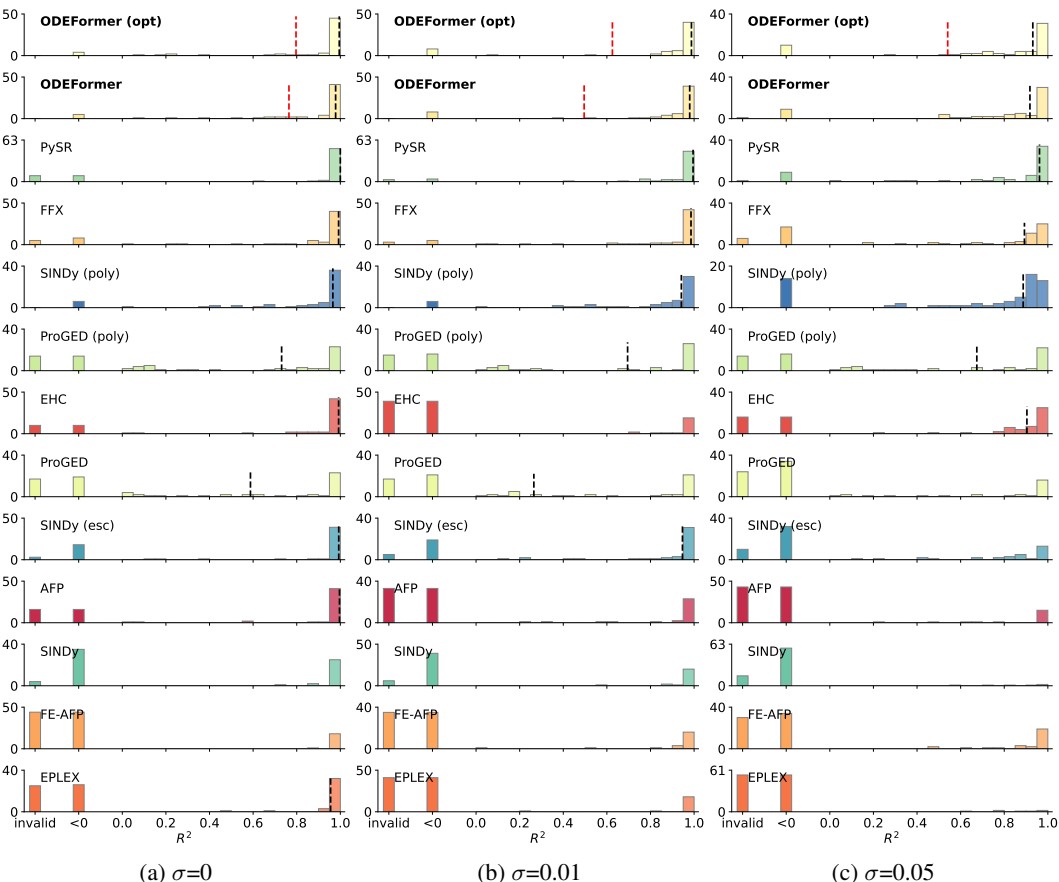

Figure 13: Histogram of per equation $R^2$ scores for the **reconstruction** task on **ODEBench**. Subfigures correspond to different noise levels. The y-axis represents counts and is scaled per model for better visibility of the distribution of scores. The x-axis annotations "invalid" and "<0" respectively denote the number of invalid predictions as well as the number of predictions that yielded an $R^2$ score below 0. The red dashed line corresponds to the mean $R^2$ score across equations, the black dashed line corresponds to the median $R^2$ score.

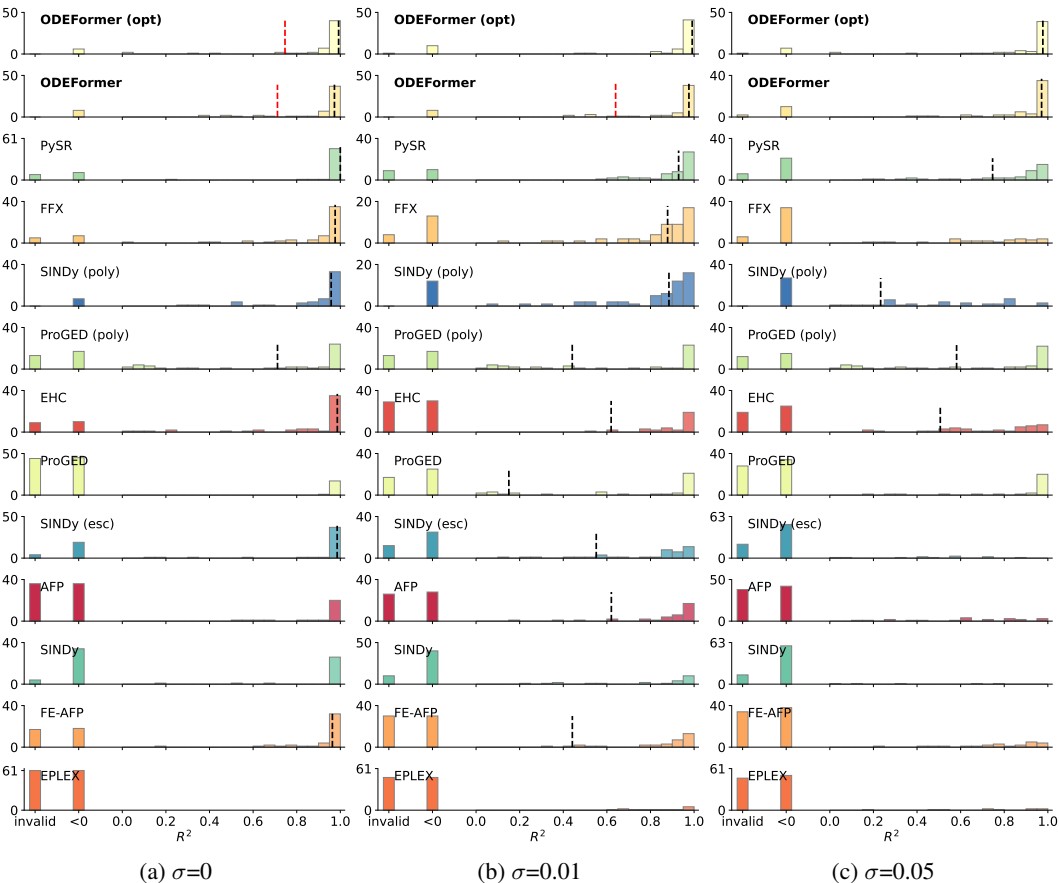

Figure 14: Histogram of per equation $R^2$ scores for the **reconstruction** task on **ODEBench** where **50%** of the trajectory are dropped uniformly at random ($\rho = 0.5$). Subfigures correspond to different noise levels. The y-axis represents counts and is scaled per model for better visibility of the distribution of scores. The x-axis annotations "invalid" and "<0" respectively denote the number of invalid predictions as well as the number of predictions that yielded an $R^2$ score below 0. The red dashed line corresponds to the mean $R^2$ score across equations, the black dashed line corresponds to the median $R^2$ score.

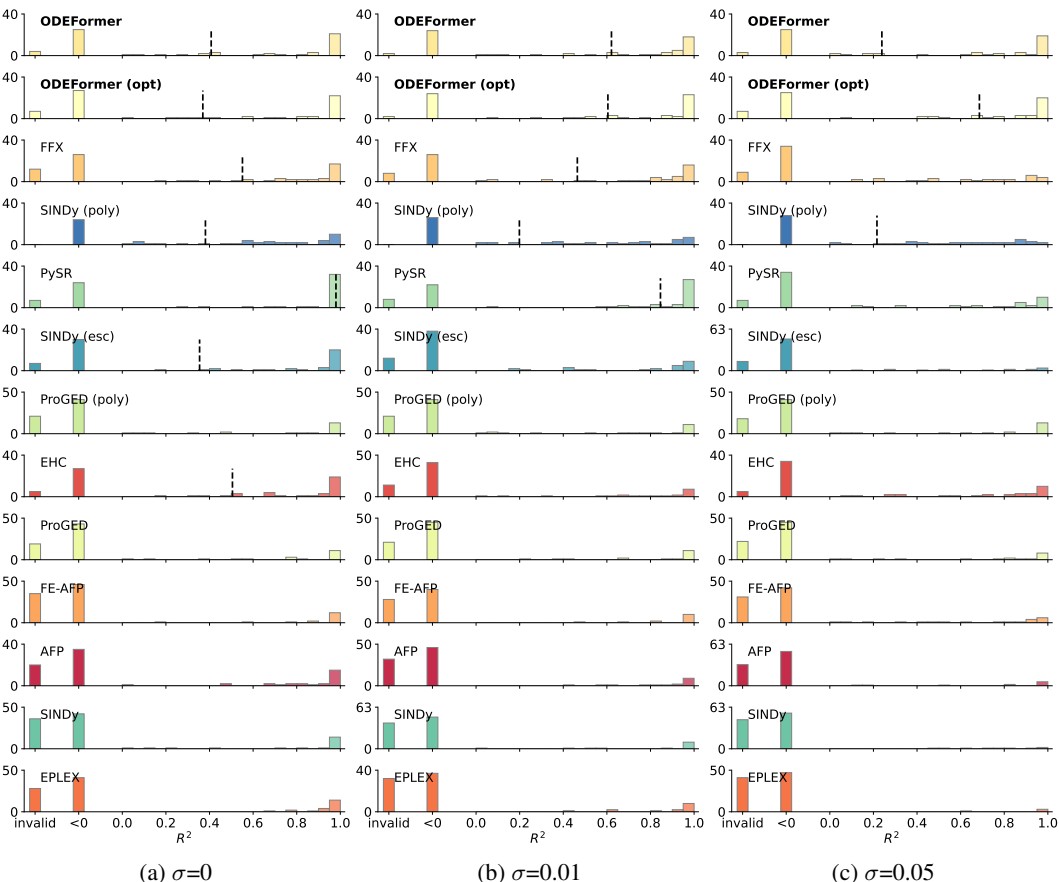

Figure 15: Histogram of per equation $R^2$ scores for the **generalization** task on **ODEBench**. Subfigures correspond to different noise levels. The y-axis represents counts and is scaled per model for better visibility of the distribution of scores. The x-axis annotations "invalid" and "<0" respectively denote the number of invalid predictions as well as the number of predictions that yielded an $R^2$ score below 0. The red dashed line corresponds to the mean $R^2$ score across equations, the black dashed line corresponds to the median $R^2$ score.

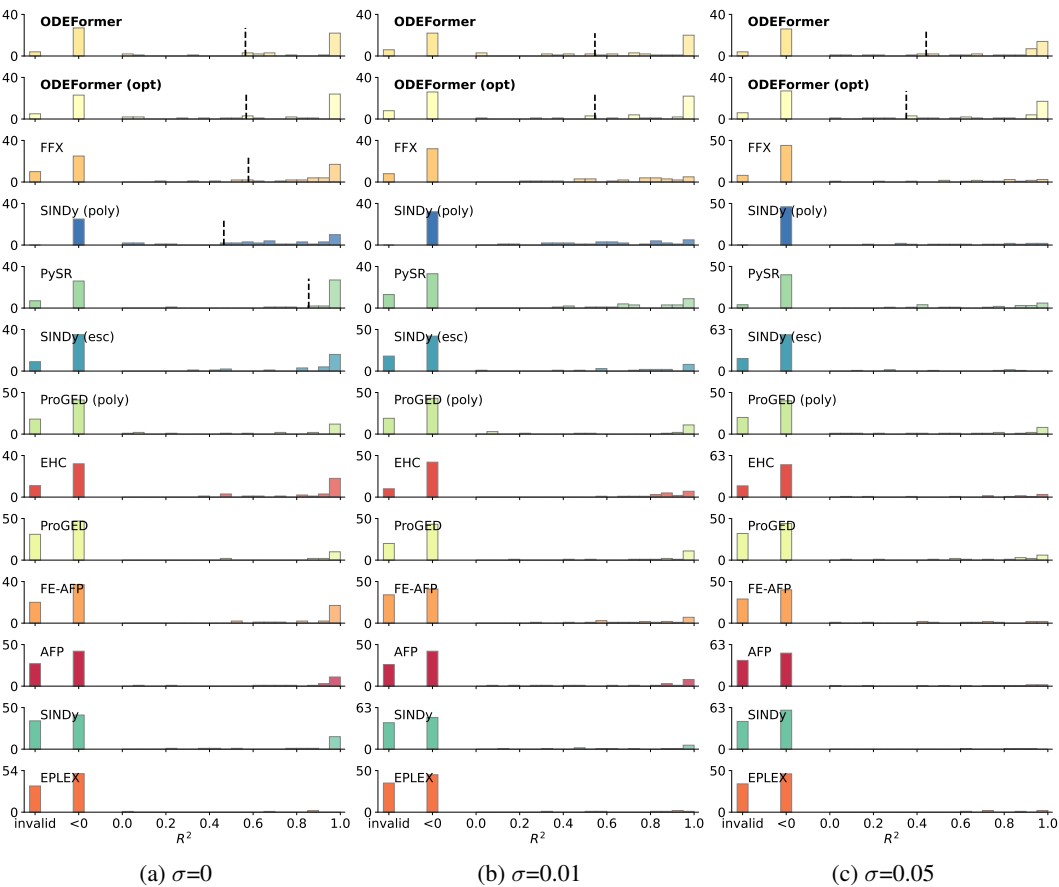

Figure 16: Histogram of per equation $R^2$ scores for the **generalization** task on **ODEBench** where **50%** of the trajectory are dropped uniformly at random ($\rho = 0.5$). Subfigures correspond to different noise levels. The y-axis represents counts and is scaled per model for better visibility of the distribution of scores. The x-axis annotations "invalid" and "<0" respectively denote the number of invalid predictions as well as the number of predictions that yielded an $R^2$ score below 0. The red dashed line corresponds to the mean $R^2$ score across equations, the black dashed line corresponds to the median $R^2$ score.

