# ODEFormer: Symbolic Regression of Dynamical Systems with Transformers

## Abstract

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

**Corrupting data.** We apply two forms of corruption to the clean solution trajectories:

- *Noise:* We sample a noise level $\sigma$ uniformly in $[0, 0.1]$ and corrupt each observation of each component of the trajectory independently multiplicatively with Gaussian noise: $x_j(t_i) \to (1 + \xi)x_j(t_i)$ for $j \in [D], i \in [N]$ and $\xi \sim \mathcal{N}(0, \sigma)$. This noise model has been used and argued for in previous works (d'Ascoli et al., 2022; Becker et al., 2023).
- *Subsampling:* For each trajectory, we sample a subsampling ratio $\rho$ uniformly in $[0, 0.5]$ and drop a fraction $\rho$ of the points along the trajectory uniformly at random. Since the equally spaced original trajectories contained between 50 and 200 points, after subsampling inputs can vary in length between 25 and 200.

---

[2]Subtractions and divisions are included via multiplication with negative numbers and the unary operator $x \mapsto x^{-1}$ respectively. It has been argued that divisions appear less frequently than additions and multiplications in "typical" expressions (Guimerà et al., 2020).

[3]This aims at avoiding deeply nested uninterpretable expressions, which often occur in GP-based SR.

[4]The oscillation of a function $h : [a, b] \to \mathbb{R}$ is given by $\sup_{x \in [a,b]} h(x) - \inf_{x \in [a,b]} f(x)$.

## 4 MODEL, TRAINING, AND INFERENCE

ODEFormer is an encoder-decoder transformer (Vaswani et al., 2017) for end-to-end dynamical SR, illustrated in Figure 2. The model comprises 16 attention heads and an embedding dimension of 512, leading to a total parameter count of 86M. As observed by Charton (2021), we find that optimal performance is achieved in an asymmetric architecture, using 4 layers in the encoder and 16 in the decoder. Since the time component is explicitly included in the inputs, we remove positional embeddings from the encoder. Model optimization follows established procedures, with details given in Appendix C.

**Tokenizing numbers.** Since numeric input trajectories as well as symbolic target sequences may contain floating point values, we need an efficient encoding scheme that allows the infinite number of floats to be sufficiently well conserved by a fixed-size vocabulary. Following Charton (2021), each number is rounded to four significant digits and disassembled into three components: sign, mantissa and exponent, each of which is represented by its own token. This tokenization scheme condenses the vocabulary size to represent floating point values to just 10203 tokens (+, −, 0, ..., 9999, E-100, ..., E100) and works well in practice despite the inherent loss of precision. We also experimented with three alternative representations: (i) two-token encoding, where the sign and mantissa are merged together, (ii) one-token encoding where sign, mantissa and exponent are all merged together, (iii) a two-hot encoding inspired by Schrittwieser et al. (2020) and used by Becker et al. (2023), which interpolates linearly between fixed, pre-set values to represent continuous values. These representations have the advantage of decreasing sequence length, and (iii) has the added benefit of increased numerical precision for the inputs. Since all three alternatives led to worse overall performance, we used the three token representation (sign, mantissa, exponent).

**Embedding numerical trajectories.** The above tokenization scheme scales the length of numerical input sequences by a factor of three. Points $(t_i, x_i) \in \mathbb{R}^{D+1}$ of the trajectory of a $D$ dimensional ODE system are hence mapped to a token sequence of dimension $\mathbb{R}^{(D+1)\times 3}$. We feed the token sequence of each dimension separately to an embedding layer and concatenate the result to obtain a representation in $\mathbb{R}^{((D+1)\times 3)\times d_{\text{emb}}}$. Before handing this representation to the encoder it is reduced such that each original input point corresponds to a single embedding vector (Kamienny et al., 2022), effectively scaling the input sequence back to its original length. For this, potentially vacant input dimensions are padded up to $D_{\max}$ before the resulting $3 \times (D_{\max} + 1) \times d_{\text{emb}}$-dimensional vector is processed by a 2-layer fully-connected feed-forward network (FFN) with Sigmoid-weighted linear unit (SiLU) activations (Elfwing et al., 2018), which projects down to dimension $d_{\text{emb}}$. This embedding process allows a single trained model to flexibly handle input trajectories of variable lengths as well as ODE systems of different dimensionalities.

**Encoding symbolic functions.** To encode mathematical expressions, the vocabulary of the decoder includes specific tokens for all operators and variables, in addition to the tokens used to represent floating point values. Importantly, the decoder is trained on expressions in prefix notations to relieve the model from predicting parentheses (Lample & Charton, 2019). With these choices, the target sequence for an exemplary ODE $f(x) = \cos(2.4242x)$ corresponds to following sequence of six tokens [cos mul + 2424 E-3 x]. For $D$-dimensional systems, we simply concatenate the encodings of the $D$ component functions, separated by a special token "|". With this simple method, the sequence length scales linearly with the dimensionality of the system, i.e., the number of variables. While this is unproblematic for small dimensions such as $D \leq D_{\max} = 6$, it may impair the scalability of our approach.[5] As the encoder is only concerned with numeric input trajectories, its vocabulary only includes tokens for numbers.

**Rescaling.** During training, the model only observes initial conditions from a standard normal distribution, and the integration range is fixed to $[1, 10]$.

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

 1. For each baseline model, we perform a separate hyperparameter optimization for each run to ensure maximal fairness. Apart from ProGED and SINDy, all baselines were developed for functional SR. We use them for dynamical SR as described in Section 2, by computing temporal derivatives $\dot{x}_i(t)$ via finite differences with hyperparameter search on the approximation order and optional use of a Savitzky-Savgol filter for smoothing. For more details, please refer to Appendix E. Note that our method is the only one which does not require any hyperparameter tuning or prior knowledge on the set of operators to be used.

Table 1: **Overview of models.** f.d.: finite differences required, ode: method developed for dynamical SR, T: transformer-based, GP: genetic programming, MC: Monte Carlo, reg: regression

| name | type | ode | f.d. | description | reference |
|------|------|-----|------|-------------|-----------|
| ODEFormer | T | yes | no | seq.-to-seq. translation | ours |
| AFP | GP | no | yes | age-fitness Pareto optimization | (Schmidt & Lipson, 2011) |
| FE-AFP | GP | no | yes | AFP with co-evolved fitness estimates | (Schmidt & Lipson, 2011) |
| EHC | GP | no | yes | AFP with epigenetic hillclimbing | (La Cava, 2016) |
| EPLEX | GP | no | yes | epsilon-lexicase selection | (La Cava et al., 2016b) |
| PySR | GP | no | yes | AutoML-Zero + simulated annealing | (Cranmer, 2023) |
| SINDy | reg | yes | yes | sparse linear regression | (Brunton et al., 2016) |
| FFX | reg | no | yes | pathwise regularized ElasticNet regression | (McConaghy, 2011) |
| ProGED | MC | yes | no | MC on probabilistic context free grammars | (Omejc et al., 2023) |

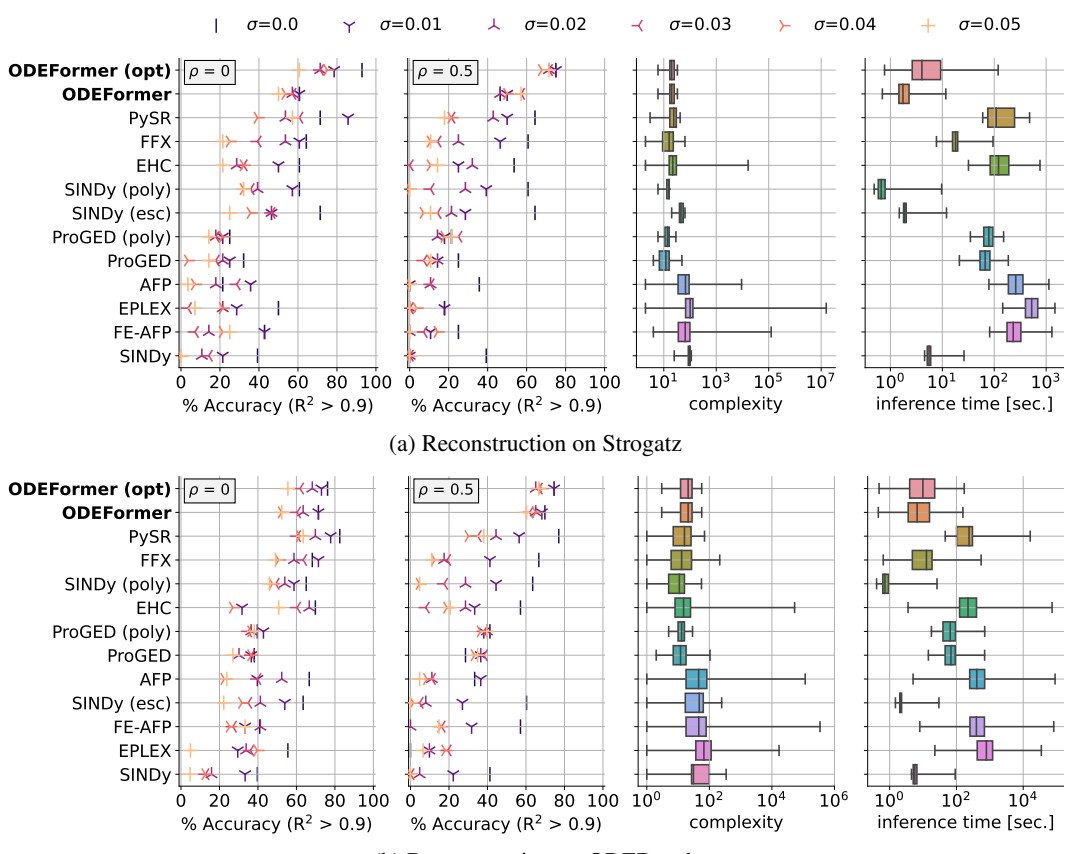

Figure 4: **Our model achieves state-of-the art performance on both benchmarks considered, while achieving higher robustness to noise and irregular sampling.** We compare ODEFormer with and without additional parameter optimization using existing methods following the protocol described in Section 6. We present results for two values of the subsampling parameter $\rho$ and six values of the noise parameter $\sigma$. Whiskers in box plot panels mark minimum and maximum values.

**Reconstruction results.** We present results on both "Strogatz" and ODEBench in Figure 4. From top to bottom, we ranked methods by their average accuracy across all noise and subsampling levels. The ranking is similar on the two benchmarks and ODEFormer achieves the highest average score on both. The two leftmost panels show that ODEFormer is only occasionally outperformed by PySR when the data is very clean – as noise and subsampling kick in, ODEFormer gains an increasingly large advantage over all other methods. In the two rightmost panels of Figure 4, we show the distributions of complexity and inference time. ODEFormer runs on the order of seconds, versus minutes for all other methods except SINDy, while maintaining relatively low and consistent equation complexity even at high noise levels. We show figures for all predictions in Appendix A.

**Generalization results.** We present generalization results on ODEBench in Figure 5. Consistently across all models, accuracies drop by about half, meaning that half the correctly reconstructed ODEs do not match the ground truth symbolically. This highlights the importance of evaluating dynamical SR on different initial conditions. Note, however, that the overall rankings of the different methods is rather consistent with the reconstruction results, and ODEFormer achieves the best results on average thanks to its robustness to noise and subsampling.

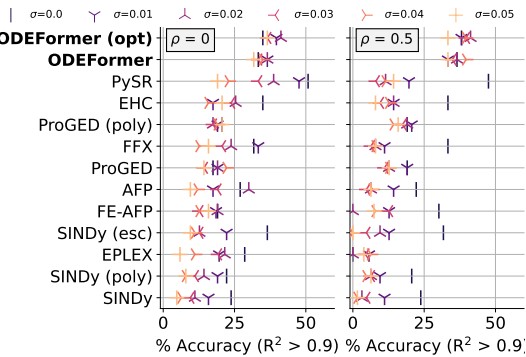

Figure 5: **Generalization on ODEBench.** We consider the same setting as in Figure 4.

# 7 DISCUSSION AND FUTURE DIRECTIONS

We presented ODEFormer, the first transformer capable of inferring multidimensional ODE systems from noisy, irregularly observed solution trajectories, as well as ODEBench, a novel benchmark dataset for dynamical SR. In extensive comparisons, we demonstrate that our model outperforms existing methods while allowing faster inference. We foresee real-world applications of ODEFormer across the sciences, for hypothesis generation of dynamical laws underlying experimental observations. However, in the following we also highlight several limitations of the current method, opening up interesting directions for future work.

First, we only considered first order ODEs. While any higher-order ODE can be written as a system of first order ODEs, this does not immediately allow ODEFormer to make predictions based only on a solution trajectory, since we would still need to approximate time derivatives up to the order of the ODE. While possible in principle via finite differencing schemes, we would suffer similar drawbacks as other methods that rely on finite differences from noisy, irregularly sampled data.