# OpenReview forum: "ODEFormer: Symbolic Regression of Dynamical Systems with Transformers"
_ICLR.cc/2024/Conference — ICLR 2024 spotlight_

### Official Review · Reviewer_hEs1 · 2023-10-30

**Soundness:** 3 good
**Presentation:** 3 good
**Contribution:** 3 good
**Rating:** 8
**Confidence:** 4

**Summary:**

The authors proposed a model for discovering the governing law of dynamical systems out of observed trajectory. The model treats trajectories as sequences of tokenized numbers and utilizes a transformer network to learn the symbolic form of governing laws from them. The authors conduct the experiment on 1D to 4D systems, and the proposed method outperforms the baseline methods.

**Strengths:**

(1) The paper is well motivated, as discovering the symbolic form of governing laws from observed data has always been the focus of scientific research.

(2) The paper contributes to neural network models for symbolic regression of ordinary differential equations.

**Weaknesses:**

(1)	The idea in this paper is not novel enough. The idea of transformer-based symbolic regression on tokenized sequences was previously reported[1]. The proposed model shares the same network structure but is applied to ordinary differential equation data. However, no analysis could be found in the paper on how the network is adapted to this new type of data.

(2)	Figures and explanations of the proposed model are way too rough and lack the necessary details.

(3)	This paper lacks crucial detail on dataset separation and the definition of tasks in the experiment section.

Reference:

[1] d'Ascoli S, Kamienny P A, Lample G, et al. Deep symbolic regression for recurrent sequences[J]. arXiv preprint arXiv:2201.04600, 2022.

**Questions:**

(1)	Could the authors give a detailed description and specific form of the loss function used during training? Is it symbolic or numerical? If the loss function is symbolic, does it measure the similarity between the symbolic tree structure or simply between the sequences? And as the ground-truth equations can be organized in different orders under both forms, which one of them is chosen in practice, and what are the reasons for refusing other possible sequences/trees?

(2)	As the authors claimed in section 2, most existing approaches of symbolic regression require a separate optimization for each new observed system. However, it is unclear in the paper whether or not the proposed model needs separate optimization on unseen systems as previous models do. Could you add some analysis on this problem and provide reasons and results to support your arguments?

(3)	The authors conduct subsampling in the experiment part and compare model performances under different subsampling rates. Could you explain further what condition in practice the subsampling process corresponds to? And how does the proposed model deal with the resulting irregular time-interval as most sequence transformers are built to encode trajectories with regular time intervals?

---

> ### Author Response · Authors · 2023-11-20
>
> Thank you for your review of our submission! We will respond to the points raised by you one by one below:
>
> ### Weaknesses
>
> 1) **Novelty**
>
> Thank you for bringing this up. There are two key differences between our proposed model and the model by D’Ascoli et al..
> The model by D'Ascoli et al.'s model is designed for recurrent sequence prediction, focusing entirely on inherently discrete sequences. An example, taken from Table 1 of D’Ascoli et al., is to find the underlying principle of the following (Josephus) sequence “0,1,1,3,1,3,5,7,1,3“, which is given by “u_n = (u_n−1 +n)%(n−1)−1”. In contrast, ODEFormer is developed to predict ordinary differential equations (ODEs), which represents a continuous function. Although input sequences for both models are discrete, ODEFormer assumes this discreteness to arise from sampling a continuous process at discrete measurement times.
>
> Furthermore, the model by D’Ascoli only considers univariate input sequences whereas our model allows for multivariate observations, i.e., it predicts the dynamic interactions between different system components. This important difference makes our model applicable to many more modeling scenarios and also manifests itself in the model architecture, in particular in the embedding layer which in our case needs to combine multiple trajectories. We have improved the description of this aspect of our model in Section 4 in order to clarify this aspect.
>
> In the revised manuscript we have added a new section in Appendix H which emphasizes the differences between our approach and the model by D’Ascoli et al..
>
> 2) **Figure and description need more details**
>
> Thank you for this feedback! In the revised manuscript we have
> - **Modified the overview figure** (Figure 2) to include more details of the proposed approach,
> - **improved the description** of the model in Section 4, and extended the section on architecture details and training hyper-parameters in Appendix C.
> We believe these adjustments indeed make the paper easier to follow and we are thus grateful for this comment. Should there still be sections that are too rough or lacking details, we would kindly ask the reviewer to point us more concretely towards them so that we can improve these.
>
>
> 3) **Experimental task and dataset separation**
>
> Thank you for this feedback! To clarify: the datasets used for the evaluation were not used for training. Indeed, this was not explicitly stated in the original submission - **we have hence updated the experimental section (Section 5)** to make this more clear. The revised manuscript also includes a concise statement of the experimental tasks.

---

> > ### Author Response · Authors · 2023-11-20
> >
> > ### Questions
> >
> > 1) **Computation of the loss function**
> >
> > Thank you for your question. We use the established encoder-decoder transformer architecture where the decoder is an autoregressive model that iteratively predicts an output sequence. In our case this output sequence is supposed to be a function (the right-hand side of an ODE), i.e., a sequence of mathematical symbols. The model is trained with the standard cross-entropy loss which, at each autoregressive prediction step, compares the predicted softmax distribution over all possible mathematical symbols vs the 1-hot distribution of the true target symbol. The loss is hence on the symbolic level and measures the similarity between the predicted vs target sequence.
> > Indeed, different symbolic sequences may correspond to the same function (e.g. 1 + x = x + 1). We do not attempt to find a canonical representation to resolve such ambiguity during training but instead use the sequence that was originally sampled. Our results in figures 4 and 5 in the manuscript show that this ambiguity does not pose a problem for training or inference in practice. Note that this ambiguity also exists in most sequence to sequence tasks: for example, there are many possible ways to translate any given sentence from one language to another, yet this does not prevent Transformers to excel at translation.
> > **To prevent future confusion, we have updated the overview figure (Figure 2) and the description of the training procedure in Appendix C.**
> >
> >
> > 2) **Is ODEFormer optimized per new observed system?**
> >
> > Thank you for your question. A separate optimization for each new observed system is *not* necessary for ODEFormer. In the submitted manuscript we allude to this in section 2 and mention it explicitly in Appendix D (“In contrast to all baseline models, ODEFormer is a pretrained model and predicted ODEs are not explicitly fit to the data observed at inference time.”). Since this is an important detail of our approach, we appreciate your feedback and agree that this aspect should be mentioned more prominently in the main text. **Therefore we have adjusted the paragraph “Model evaluation” in Section 5 of the revised manuscript**, which now includes the following:
> >
> > “All baseline models are fitted to each equation separately, including a separate hyperparameter optimization per equation. In contrast, we evaluate a single, pre-trained ODEFormer model on all equations in all benchmarks.”
> >
> > Importantly, although our model does not require any optimization per newly observed system, it is optionally possible to finetune the constants of a predicted ODE to the particular observed system. This step is described in section 4 “Optional parameter optimization”. In all of our experiments (section 6) we show results of the fully end-to-end ODEFormer as well as ODEFormer with post-hoc constant fine-tuning, dubbed ODEFormer (opt). While on Strogatz, the additional tuning considerably increases accuracy, the effect is less pronounced on ODEBench and for the generalization results (Figure 5).  Despite the additional constant tuning, ODEFormer (opt) remains one of the fastest models in the comparison. In the revised manuscript we added an explicit statement of these results.

---

> > > ### Author Response · Authors · 2023-11-20
> > >
> > > 3) **Randomly subsampling trajectories**
> > >
> > > Thank you for this question. Subsampling simulates measurement processes that do not follow a precise sampling grid, which is a common property of time series data. Examples can be found in natural science but also in many other domains:
> > >
> > > - Biology: The Lotka-Volterra equations are a famous ODE system from ecology and describe the interaction of two populations of species, typically predator and prey. Fitting this model to data requires empirical observations of the two populations (e.g. counting how many species there are on a given day). In practice such observations may often not follow a perfect sampling grid e.g. due to weather conditions, logistical constraints on the field study, or perhaps simply because species were only counted on weekdays but not on the weekend. Instead there will be missing observations, i.e. the sampling grid is irregular.
> > > - Physics: A second example of empirical observations obtained on an irregular sampling grid is given by particle collision data in high-energy physics experiments, like those conducted at the Large Hadron Collider. The detection of subatomic particles resulting from high-energy collisions occurs at irregular intervals, dictated by the random nature of the particle collisions and the intervals at which these collisions are recorded by detectors. This results in datasets that are inherently irregular.
> > > - Healthcare Monitoring: In patient health monitoring, data points might be recorded irregularly based on the patient's condition. For instance, heart rate or blood sugar levels might be monitored more frequently during a medical event or change in condition, but less frequently otherwise.
> > > - Financial Markets: In finance, transactions or market events do not occur at regular intervals. Stock prices, for example, are recorded at the time of each trade, which can vary substantially throughout the trading day.
> > >
> > > Irregular sampling grids (in particular in combination with additional measurement noise) pose a severe problem for models that rely on finite difference estimates of temporal derivatives, which become inaccurate under signal corruptions. However, in contrast to almost all baseline models (see table 1), ODEFormer does not require finite difference estimates. Instead we include the time information itself as part of the input that is supplied to the transformer (see section 4: “Embedding numerical trajectories” ). Since we train the model on corrupted (noisy and randomly subsampled) data, the model learns to become invariant with respect to irregular sampling grids. **We have updated the manuscript to make this aspect of our approach more clear.**

---

> > ### Comment · Reviewer_hEs1 · 2023-11-22
> > **reply to response**
> >
> > -**On the novelty of the proposed model.** I can now understand the difference of the task between your work and the previous paper. However, it would seem to me that despite different tasks, the   previously reported model needs no adjustment to apply on the new task, particularly with the very similar method proposed by the authors. Could you explain further the difference between the previous model and yours from the methodology perspective and how exactly does the difference between sequences and ODEs affect the design of your model?
> >
> > -**The adjustments in figures, descriptions and experimental details.** Thank you for the update and for taking my suggestions into account.
> >
> > -**Computation of the loss function.** Thank you for the detailed explanation. I find your argument about the ambiguity of symbolic sequences quite convincing.
> >
> > -**The optimization.** Thank you for the adjustment. It would make the paper friendlier to the readers.
> >
> > -**The subsampling issue.** It is quite clear in the explanation the corresponding real-world scenarios. As you have pointed out, the irregular sampling grids would definitely harm those methods using finite difference estimates. However, it is unclear about the reason of the proposed model outperforming those methods without finite difference. Could you mark in the baselines those with finite difference estimates and those without, and add some analysis on the advantages of your model against other methods free of finite difference? Meanwhile, I have just noticed that the baseline models lack some crucial methods proposed recently such as [2].
> >
> > **Overall** The paper still needs more analysis on the method itself about its distinction over the previous model and the reason of its performance. Some crucial baseline models proposed recently should be added as well. Nevertheless, the paper is now clear in representation and contains detailed analysis on some vital problems. I would increase my score if proper analysis from the methodology perspective can be added as well as more recent baselines.
> >
> > Reference:
> > [2] Qian, Zhaozhi, Krzysztof Kacprzyk, and Mihaela van der Schaar. "D-code: Discovering closed-form odes from observed trajectories." International Conference on Learning Representations. 2022.

---

> > > ### Author Response · Authors · 2023-11-22
> > >
> > > Thank you for your detailed reply and for helping us further improve our submission. Regarding the remaining questions and issues:
> > >
> > > **1) On the novelty of the proposed model.**
> > >
> > > Thank you for acknowledging the improvement and helping us to further clarify this point. We will answer in multiple parts.
> > >
> > >
> > > > “However, it would seem to me that despite different tasks, the previously reported model needs no adjustment to apply on the new task"
> > >
> > > Let us consider a concrete example for our reply here, e.g., the simple ODE $\dot{x}(t) = -x(t)$. The solution to this ODE is $x(t) = x_0 \exp(-t)$. Assuming the initial condition $x(0) = x_0 = 1$ we can simulate observational data via
> > > ```python
> > > time = np.linspace(0, 10, 10)
> > > observations = np.exp(-time)
> > > # observations = [1.00000000e+00, 3.29192988e-01, 1.08368023e-01, 3.56739933e-02, 1.17436285e-02, 3.86592014e-03, 1.27263380e-03, 4.18942123e-04, 1.37912809e-04, 4.53999298e-05]
> > > ```
> > > Indeed, this is a list of numbers that we could plug as input into the model by D’Ascoli et al.. While the trained model is not publicly available, there is an online demo (https://symbolicregression.metademolab.com/), which allows one to run the model on sequences of up to 30 numbers. For the above input, the model predicts the symbolic recurrence relation $u_n = \exp\left(\frac{10}{9} - \frac{10 n}{9}\right)$, so indeed, as you were contemplating, we do get a valid prediction from their model.
> > >
> > >
> > > However, based on this example, we make the following general observations about differences to our model:
> > > - A *first key limitation* is that their model only works on scalar input sequences. The extension to multi-variate systems is one of the key novelties of our paper.
> > > - The model of D’Ascoli et al. always interprets the list of provided numbers as a contiguous, integer-indexed sequence $u_n$. One might be tempted (especially in our regularly sampled, noiseless example) to simply rewrite the recurrence relation $u_n$ as a continuous function of time instead, i.e., $u(t) = \exp\left(\frac{10}{9} - \frac{10 t}{9}\right)$, which on the skeleton level reads $\exp(c - c \cdot t)$. Ignoring the constants, the functional form of the prediction resembles the function of the solution trajectory. However, this implies that their model predicts an analytical expression of the *solution to the ODE* whereas we are interested in predicting the ODE itself (the correct answer would hence be $\dot{u}(t) = -u(t)$. This illustrates that this model was developed for a different use case and can not be directly applied in our setting (and hence also not be included in the benchmarking experiments.)
> > > - We iterate that their model expects integer-indexed contiguous values as inputs. For example, had we sampled the solution trajectory in the example above at different time values, their model would always predict another recurrence relation, whereas our model would provide the same output.
> > >
> > > **As such we can answer your question in the negative: The model by D’Ascoli et al. can *not* be applied in our scenario without adjustment.**
> > >
> > >
> > > > “Could you explain further the difference between the previous model and yours from the methodology perspective and how exactly does the difference between sequences and ODEs affect the design of your model?”
> > >
> > > On the concrete methodological level, the key difference stems from the fact that we consider multi-dimensional input sequences, which the previous model cannot handle. This is a substantial extension and primarily enabled by how we embed the tokenized input sequences (more precisely, in case of the model by D’Ascoli et al. there is only a single token sequence whereas our model needs to combine one token sequence per dimension + one token sequence representing sampling times). We have provided more details on this in the updated overview Figure 2, as well as in the paragraph on “Embedding observed trajectories” in Section 4.
> > >
> > > Since on an abstract level, both our work as well as D’Ascoli et al. solve a sequence-to-sequence task with sequences of float numbers as input and sequences of operators and symbols as output, the overall encoder-decoder transformer architecture is indeed quite similar.
> > >
> > > However, we believe the similarities in model architecture should not be viewed as a lack of novelty. The major aspect of novelty lies in the task we’re solving, which for instance also largely depends on how we generate, filter, and represent the data which the seq2seq model is ultimately trained on.
> > > To this end, we need to emphasize the data generation procedure for such a dataset, which we describe in detail in Section 3. We will also release our training dataset in addition to the model itself. In addition, we also provide the most carefully curated and comprehensive benchmark dataset for this task to date. (While not marking a difference between the models, this constitutes a relevant aspect of the novelty.)

---

> > > > ### Author Response · Authors · 2023-11-22
> > > >
> > > > **2) The subsampling issue.**
> > > >
> > > >
> > > > > It is quite clear in the explanation the corresponding real-world scenarios.
> > > >
> > > > We are glad that we could clarify this issue and are happy to expand on your questions here.
> > > >
> > > > > As you have pointed out, the irregular sampling grids would definitely harm those methods using finite difference estimates. However, it is unclear about the reason of the proposed model outperforming those methods without finite difference. Could you mark in the baselines those with finite difference estimates and those without [...],
> > > >
> > > > Thank you for this suggestion. For a distinction between models using finite differencing vs. those that do not, please refer to table 1 which includes this information.
> > > >
> > > >
> > > > > [...] and add some analysis on the advantages of your model against other methods free of finite difference?
> > > >
> > > > Thank you for the suggestion of comparing in more detail to other finite-difference-free methods to get a better understanding of the corresponding benefits. As shown in table 1 of our manuscript, the only other model not using finite difference approximation is ProGED. ProGED operates in two steps:
> > > > - According to a user-defined probabilistic grammar, it firstly generates a candidate set of “skeleton functions”, where by skeleton function we mean a function with placeholders instead of numerical constants, e.g. the skeleton of 2*x+1 would be C*x+C.
> > > > - These placeholder constants are fitted to the observations in step two.
> > > >
> > > > Upon taking a closer look at the implementation of step two (constant fitting), we were surprised to notice that ProGED actually also uses finite differences, citing from their manuscript (Section 4.2.1, page 8, end of second paragraph):
> > > >
> > > > *“The objective function for the optimization algorithm was the root-mean-square error between observed (**numerically calculated derivatives**) and predicted values of time derivatives of the state variables”*.
> > > >
> > > > We previously missed this detail as the authors earlier, when introducing their model (section 3.2, page 5, first paragraph), write
> > > >
> > > > *“**To address the limitations of numerical differentiation**, we introduce an approach based on simulating differential equations.”*,
> > > >
> > > > which we interpreted as not requiring finite difference approximations. Moreover, their model implementation does not require us to supply finite difference approximations (presumably as the model estimates them internally) which seemed to confirm our initial (incorrect) understanding that the model is free of finite-differencing approximations.
> > > >
> > > >
> > > > This leads us to the (unexpected) conclusion that ODEFormer is in fact the only model not using finite differences - which may in turn explain its superior robustness to signal corruptions. We have updated table 1 accordingly and would like to thank you again for encouraging us to further investigate this issue, which helped us to spot this error.

---

> > > > > ### Author Response · Authors · 2023-11-22
> > > > >
> > > > > >Meanwhile, I have just noticed that the baseline models lack some crucial methods proposed recently such as [2].
> > > > > We are glad that we could clarify this issue and are happy to expand on your questions here.
> > > > >
> > > > >
> > > > > Thank you for pointing us to this reference, which we indeed missed to include in our original submission. We have already added it as a reference in the related work section in the revised manuscript and are working on including it in the evaluation.
> > > > >
> > > > >
> > > > > Regarding the inclusion of additional baselines in our benchmark evaluation, we would like to offer the following considerations:
> > > > > In addition to the eight baseline models that we already include in our evaluation (not counting multiple configurations for SINDy and ProGED), we are aware of the following papers for symbolic regression for ODEs:
> > > > > - Gaucel et al., 2014, “Learning Dynamical Systems Using Standard Symbolic Regression”
> > > > > - La Cava et al., 2016a, “Inference of compact nonlinear dynamic models by epigenetic local Search”
> > > > > - Quade et al., 2016, “Prediction of Dynamical Systems by Symbolic Regression”
> > > > > - Atkinson et al., 2019, “Data-driven discovery of free-form governing differential equations”
> > > > > - Weilbach et al., 2021, “Inferring the Structure of Ordinary Differential Equations”
> > > > > - Kronenberger et al., 2021, “Identification of Dynamical Systems using Symbolic Regression”
> > > > >
> > > > >
> > > > > We cite all of these in our manuscript, however, unfortunately, there is no implementation available for any of these models. As such we were unable to include them in the evaluation. In fact, the lack of available implementations led us to include many of the baseline models originally developed for functional symbolic regression in our benchmark in the first place (that is models for “standard” symbolic regression for algebraic equations, not ODEs; this distinction is also included in table 1 of our manuscript).
> > > > >
> > > > >
> > > > > We are also aware that there are many more models available for functional symbolic regression (and we cite many of them in the related work section), however, applying them to ODEs would in all cases require finite difference approximations, implying the same lack of robustness witnessed in our experiments (which we seem to agree on). Moreover, the models we include in the evaluation already cover the major classes of approaches (evolutionary algorithms, regression based models, probabilistic grammar/Monte Carlo approaches). While we agree that it would be great to also include the reinforcement based approach from Petersen et al., 2020 (often referred to as “DSR”; already included in related work section) and the hybrid approach by Udrescu et al. 2020 (known as “AIFeynman”, also included in related work) as these might be regarded as alternative approaches to functional symbolic regression, these models are unfortunately known to be inherently slow (their median inference time per equation is reported to be $10^4$ seconds (= 2.77 hours) in the benchmark study by LaCava et al., 2021 (figure 1)). An evaluation on the 28 examples of 2-dimensional systems across 6 noise levels and 2 subsampling levels would hence amount to approximately
> > > > > $(10^4) \cdot 2 \cdot 28 \cdot 6 \cdot 2 / (60 \cdot 60 \cdot 24) = 77.7$ days of compute time, which was simply infeasible for us.
> > > > > Furthermore, since both DSR and AIFeynman would also require finite difference approximations when applied to ODEs, we do not see reason to believe that they would offer increased robustness to noise.
> > > > >
> > > > > > “[inclusion of] more recent baselines”
> > > > >
> > > > > Lastly, with regard to your suggestion to also include “more recent baselines”, we would like to point out that with PySR and ProGED we already include two models that were only recently published (both in 2023).
> > > > >
> > > > > **Nevertheless, we are open to include more baseline models in the final version of the paper in case you feel that we missed a major aspect in our considerations.**

---

> > > > > > ### Author Response · Authors · 2023-11-23
> > > > > >
> > > > > > Thank you again for pointing us the paper by Qian et al.. We are working on the evaluation of this model but unfortunately face some issues with the implementations which is why we can not yet update the manuscript with the additional results in figures 4 and 5. Specifically, there seems to be no simple way to evaluate this model on new datasets (the code is very specific to the exact datasets and hyper-parameters used in the paper, e.g. every dataset has its own class (with dataset specific properties, see here https://github.com/ZhaozhiQIAN/D-CODE-ICLR-2022/blob/main/equations.py#L263:L878), model configuration (https://github.com/ZhaozhiQIAN/D-CODE-ICLR-2022/blob/main/config.py#L115:L267) as well as interpolation configuration (https://github.com/ZhaozhiQIAN/D-CODE-ICLR-2022/blob/main/config.py#L52:L113). Unfortunately there is no documentation available to this code, which makes it hard to understand how to select hyper-parameters correctly or even how to run the model on all component functions of an ODE system. As a result, we currently mostly obtain ‘NaN’ predictions or ODEs which are simply $f(x) = 0$.
> > > > > >
> > > > > >
> > > > > > We will reach out to the authors and ask for some clarification on how to use this model. In the meantime we also noted that this model seemingly requires a lot of tuning to individual equations, e.g. there seems to be a new set of hyper-parameters per ODE system. For instance, the set of admissible operators as well as the admissible range for constants differ between systems. Although we like the theoretical framework proposed in this paper, access to this kind of knowledge seems unrealistic to us in practice. Nevertheless, we will do our best to include this model in the final version of our paper.

---

> > > > > > > ### Comment · Reviewer_hEs1 · 2023-11-23
> > > > > > >
> > > > > > > Thanks for the detailed reply and for taking my suggestions into account.
> > > > > > >
> > > > > > > I now understand that the novelty of your proposed model mainly lies in discovering the ODE invariant of the time grid rather than a single sequence. As for the baselines, since the authors have contained some recent methods while the remaining ones lack the necessary documents for quick repetition, I agree that the current baseline models are sufficient for comparison.
> > > > > > >
> > > > > > > Based on the changes have made and your replies, I will update my score.

---

### Official Review · Reviewer_znUh · 2023-11-01

**Soundness:** 3 good
**Presentation:** 2 fair
**Contribution:** 1 poor
**Rating:** 3
**Confidence:** 3

**Summary:**

ODEFormer, a transformer model designed for dynamical symbolic regression. This model is capable of inferring multidimensional ordinary differential equation (ODE) systems from noisy and irregularly sampled data. It utilizes a pre-trained sequence-to-sequence transformer on synthetic data to generate symbolic expressions directly from observations. The paper also introduces ODEBench, a benchmark dataset for dynamical symbolic regression, comprising 63 ODEs sourced from the literature, modeling real-world phenomena across dimensions one to four, including chaotic systems.

The reviewer further highlights the evaluation and comparison of ODEFormer with existing methods. It assesses ODEFormer's performance on both the established Strogatz dataset and the new ODEBench dataset. The comparison encompasses various techniques based on genetic programming, regression, and Monte Carlo methods.

**Strengths:**

The introduction of ODEFormer represents a novel framework that serves the purpose of generating ordinary differential equations (ODEs) specifically designed for testing dynamical systems. This innovative approach allows researchers and practitioners to create ODE models that can accurately capture and represent the dynamics of real-world systems, providing a valuable tool for testing and understanding the behavior of complex dynamical systems.

ODEFormer introduces a pioneering use case for transformers in the realm of ODEs. Traditionally, transformers are employed in natural language processing and sequential data tasks. However, in this context, they are harnessed to directly infer ODE systems from noisy and irregularly sampled data. This expansion of transformer applications into the domain of ODEs signifies a breakthrough, offering a versatile and data-driven approach for modeling and analyzing complex dynamic systems, and opening up new possibilities for the fusion of machine learning techniques with physics-based modeling.

**Weaknesses:**

One notable limitation in the presented work is the absence of comparisons with benchmarks employed in previous studies, such as the widely recognized benchmark datasets used in Neural ODE (Chen et al). This absence makes it challenging to gauge how the proposed ODEFormer framework performs in comparison to existing approaches on well-established and widely accepted testing scenarios.

The demonstrated applicability of ODEFormer on toy datasets represents another potential limitation. Toy datasets are typically simplistic and may not fully capture the complexity and variability encountered in real-world applications. Therefore, the extent to which ODEFormer can effectively handle and model more complex, real-world data remains an open question and warrants further investigation and validation on diverse and challenging datasets.

**Questions:**

One key aspect that warrants further exploration is the practical application of the technique on real-world datasets. While the framework shows promise in artificially constructed datasets, its effectiveness in solving real-world problems, where data can be noisy, irregularly sampled, and complex, remains to be demonstrated. Evaluating its performance on non-artificial, real-world datasets across various domains, such as finance, healthcare, or environmental monitoring, would provide valuable insights into its applicability and limitations in practical scenarios.

Time series data indeed represents a compelling use case for the framework. Time series forecasting is a critical application in various fields, including finance, energy, and climate modeling. Therefore, it is crucial to investigate the applicability of this method in a time series forecasting modeling scenario. Demonstrating its effectiveness in accurately modeling and predicting time-dependent data can significantly enhance its practical utility and establish its relevance in solving real-world, dynamic data challenges.

---

> ### Author Response · Authors · 2023-11-21
>
> We thank the reviewer for their time.
>
> There are several aspects of the review on which we respectfully disagree with:
>
> **1) The claim that we did not compare our method with existing approaches on established datasets:**
>
> In fact, the bulk of the paper consists of such comparisons (to relevant baselines such as Sindy, etc., as well as on established datasets such as the ‘Strogatz’ dataset which has been used in numerous previous works, e.g.  [1, 2, 3, 4]). Oddly, this aspect of our submission is also emphasized in your summary:
> “The reviewer further highlights the evaluation and comparison of ODEFormer with existing methods. It assesses ODEFormer's performance on both the established Strogatz dataset [...]”
>
> References:
>
> - [1]: Nina Omejc et al. 2023, “Probabilistic grammars for modeling dynamical systems from coarse, noisy, and partial data”
> - [2]: Kronenberger et al. 2021, “Identification of Dynamical Systems using Symbolic Regression”
> - [3]: La Cava et al. 2021, “Contemporary Symbolic Regression Methods and their Relative Performance”
> - [4]: La Cava et al. 2016, “Inference of Compact Nonlinear Dynamic Models by Epigenetic Local Search”
>
>
> **2) The claim that we only consider toy datasets without noise and irregular sampling:**
>
> A major advantage of our approach, emphasized throughout the paper, is that our method is much more robust to noise than existing methods, as demonstrated empirically. Our experiments in particular already consider noisy and irregularly sampled data.
>
>
> **3) The suggestion to include the established dataset from the Neural ODE paper:**
>
> Having inspected the referenced Neural ODE paper again, we are unsure which benchmark dataset the reviewer is referring to as this reference simply does not offer any benchmark datasets. Moreover, as discussed in the related works section, we would like to emphasize that Neural ODEs operate in a completely different setting to ours: Neural ODEs are black box models which do not output a symbolic expression. Hence, it has higher expressivity, but is void of (symbolic) interpretability.
>
> **4)** We are also surprised that although the general tone sounds very positive as exemplified by your summary sentences below, the score is very negative.
> - "This **innovative** approach […] providing a **valuable tool** for testing and understanding the behavior of complex dynamical systems”
> - "ODEFormer introduces a **pioneering use case** [...].“
> - "This expansion of transformer applications into the domain of ODEs **signifies a breakthrough**, offering a versatile and data-driven approach for modeling and analyzing complex dynamic systems, and **opening up new possibilities** for the fusion of machine learning techniques with physics-based modeling.”
>
>
> In particular, we would like to ask the reviewer how this summary of our submission leads to a contribution score of 1 (poor).
>
>
> **In light of these contradictions, it is hard for us to offer a sound rebuttal and a constructive revision that improves our submission.**

---

### Official Review · Reviewer_hbTD · 2023-11-01

**Soundness:** 4 excellent
**Presentation:** 4 excellent
**Contribution:** 4 excellent
**Rating:** 8
**Confidence:** 4

**Summary:**

The paper studies the problem of symbolic regression for dynamical systems, specifically ODE, with the use of transformers. Apart from the adjustments need to use a transformer based model for this task, authors also propose a new benchmark. This is claimed to be more diverse and larger than existing ones. Empirically, the proposed method is shown to outperform existing baseline methods in terms of reconstruction as well as generalization.

**Strengths:**

The paper is extremely well-written (including notations, clearly stating contributions, etc.), very well motivated and the proposed method is shown to achieve state of the art performance.

The placement within existing literature is very well articulated.

Since, authors propose a benchmark, it is appreciated that the data generation procedure is outlined precisely.

The section on filtering of the data to avoid rapidly converging and divergent systems is worth mentioning, details like these make a benchmark standout.

Tokenization, embedding process and encoding of the symbolic functions is very well justified.

Baseline methods have been chosen appropriately and the empirical evidence is very convincing.

**Weaknesses:**

The authors have mentioned a few of their limitations, which is great.

While authors mention the presence of a very related work "Becker, Sören, et al. "Predicting Ordinary Differential Equations with Transformers." (2023).", I am not sure why they do not elucidate the difference between the paper mentioned and their proposed method, why is this method not used as a baseline? There is no attempt to illustrate the difference at all.

I have another point which I would like to bring up with the authors regarding the generation of the data, specifically the way it is being integrated. As mentioned (and I think this is reasonable as a first step) the authors using fixed homogeneous grid to integrate, and it is claimed that number of points don't matter during inference (Figure 3), is this indeed an artifact of the way the data is generated. How do the results change when the integration procedure is altered, it would be great to have the authors comment on this issue.

Please see questions section for further.

**Questions:**

1) Please compare and contrast the proposed method with the work "Becker, Sören, et al. "Predicting Ordinary Differential Equations with Transformers." (2023).". As acknowledged that this is closely related, it should be clearly articulated what are the differences. The mentioning of difference between univariate and multivariate is okay, but this needs more explanation. I strongly believe this should be a baseline for comparison in some setting.

2) How should one think about inferring PDE with a variant of this framework, can we have a discussion around this as part of limitation if that is the case?

3) The learning from multiple trajectories, doesn't seem to give promising performance, do authors have comments on explaining this behavior?

4) For the noise corruption of the data, the authors have tried out adding Gaussian noise and dropping samples uniformly at random. Can authors comment on missing chunks of data instead. This might be of practical implication in certain cases, where the data collection mechanism (sensor) went faulty for some time before resuming activity.

**Details Of Ethics Concerns:**

Authors have very aptly mentioned the limits that could arise in the very last paragraph of the paper, which is very much appreciated. I don't believe further review is needed.

---

> ### Author Response · Authors · 2023-11-20
>
> Thank you for your review of our submission! We will respond to the points raised by you one by one below:
> ### Weaknesses
> 1) **Comparison to Becker et al. 2023**
>
> Thank you for raising this issue. We noted the major difference in the related works (Section 2): the model proposed by Becker et al. (2023) can only be applied to univariate ODEs, while our model can handle multivariate ODEs. However, we agree that this comparison deserves more details and **added a full section (“DETAILED COMPARISON WITH NSODE (Becker et al. (2023)”) in Appendix H** in the revised manuscript, describing the similarities and differences w.r.t. Becker et al. (2023).
>
> Since the model by Becker et al. (2023) can only be applied to univariate ODEs, it can not be benchmarked on the multivariate datasets used in our paper (Strogatz, ODEBench). Instead **we added an evaluation and performance comparison between ODEFormer and Becker et al. (2023)** using the original 1D datasets suggested by Becker et al. (2023). Our results show that ODEFormer shows competitive performances on those datasets as well, for details please refer to Appendix H.2 in the revised manuscript.
>
> 2) **Question on robustness wrt ODE integration**
>
> Thank you for bringing up this point, which reveals that our formulation was misleading: to clarify, we do not integrate on a homogeneous grid. Instead, we use the Runge-Kutta method of order 5(4) (often called “rk45”) which is a classic adaptive step size solver. At every integration step, adaptive step size solvers choose the maximally allowed step size, which still keeps the integration error below a user-defined tolerance threshold.
> In practice, this usually results in small steps within areas of high variation and larger steps in more “flat” areas, yielding an irregular sampling grid. While adapting the step size greatly increases integration speed,  the irregular sampling is often undesirable for modeling real-world data.
>
>  In our particular case, the irregularities would be informative of the signal variation - an unreasonable general assumption for naturally observed data. Fortunately, most numerical solvers (in particular the rk45 implementation underlying scipy.integrate.solve_ivp, which we use) accept an additional parameter ‘time’ which allows the user to define a time grid along which one would like to obtain the numerical solution. In this case, the final solution returned by the solver corresponds to an interpolation of the adaptive step size solution, evaluated on the user-defined time grid.
>
> In our data generation procedure (Section 3), we supply a fixed homogeneous grid of N points, where N is sampled independently for each generated sample from a discrete uniform distribution, i.e. N ~ U{50, 200}. Since a homogenous grid would yet again correspond to an unreasonable assumption for naturally observed data, we simulate irregular sampling by dropping a fraction of p time points at random (as described in Section 5 “Corruptions”).
> So while the internal adaptive solver uses irregular steps, the numerical solution is an interpolation on a  homogeneous grid from which we randomly remove points to simulate imperfect real-world sampling. We believe this sampling procedure to be sound.
>
> We have clarified this sampling procedure in the revised manuscript to avoid potential confusion.
>
> Nevertheless, you bring up an interesting question regarding the robustness of ODEFormer w.r.t. the number of time points in an observation. The datasets in the additional comparison with Becker et al. (2023) (Appendix H in the revised manuscript, see our previous comment) include trajectories with 256 timepoints, which exceeds what ODEFormer has seen during training (trajectories during training include between 25 to 200 timepoints). Across all three datasets, results on trajectories with 256 time points are similar to results (frequently even exceeding those) with 128 time points, giving evidence that the robustness with respect to the number of time points observed in Figure 3 is not strictly limited to the training distribution.
>
> In our original manuscript we write “[...] ODEFormer is surprisingly insensitive to the number of points in the trajectory” (section 5, “Results on synthetic data”). We did not intend to “claim that the number of points don't matter during inference” but instead that one intuitively might expect better results when more data is available (hence the surprise). However, **we can see how our statement can lead to your interpretation and have hence rephrased this result in the revised manuscript to avoid confusion.**

---

> > ### Author Response · Authors · 2023-11-20
> >
> > ### Questions:
> >
> > 1) **Comparison to Becker et al.**
> >
> > Please see our first comment above.
> >
> >
> > 2) **Possible extensions to PDEs**
> >
> > Thank you for this suggestion! We agree that extensions to other classes of differential equations, e.g. partial differential equations (PDEs) or stochastic differential equations (SDEs), seem like an exciting direction for future work. A potentially limiting factor may, however, be the difficulty of integrating PDEs numerically with sufficient accuracy and speed to generate a large-scale training dataset. Since we propose a model for ordinary differential equations (ODEs), we do not see potential problems on PDEs as a limitation of our current model. Inspired by your suggestion, we instead **updated the manuscript and added a brief discussion of potential extensions to PDEs and associated concerns** in the conclusion in section 6 (“Conclusion”).
> >
> >
> > 3) **Inference from multiple trajectories**
> >
> > In the presented model there is currently no principled way to perform inference from multiple trajectories as this was not our original intent. A naive approach for inference from multiple observations of the systems is to collect these in a minibatch which is passed to the model and to average the decoder logits across the minibatch. Empirically, this approach often resulted in invalid mathematical expressions. A potential explanation may be that naive logit averaging can boost the probability for tokens that would not have been selected for any of the individual observations in the minibatch. As such, inference from multiple trajectories remains an open issue, as mentioned in the section on model limitations. Despite this limitation, though, we would like to emphasize that it can also be regarded as a principled strength that our model does not require multiple trajectories of the same system.
> >
> >
> > 4) **Missing data chunks**
> >
> > Thank you for this great question! Indeed, this seems like a relevant practical use-case. **We have added an experiment with missing data chunks in appendix F** in the revised manuscript. In summary, we assess the performance of ODEFormer for three different chunk sizes and show that the model retains reasonably robust performance across all tested settings. For details please refer to appendix F. Thank you for the idea of this additional experiment!

---

> > > ### Comment · Reviewer_hbTD · 2023-11-21
> > >
> > > Dear authors,
> > >
> > > Thank you for the detailed rebuttal. I am glad that the comments were helpful, hopefully these will get reflected in the final version. Since, my questions have been answered, I am updating my score accordingly.
> > >
> > > Thank you

---

### Author Response · Authors · 2023-11-20

We thank all reviewers and the Area Chair for their work on our submission. We have uploaded a revised manuscript implementing the reviewers suggestions. Sections that have changed compared to the original submission are marked in orange. We will address individual points raised in the reviews in separate responses to the reviewers below.

---

### Meta-Review · Area_Chair_8NEv · 2023-12-05

**Metareview:**

This paper proposes to use a encoder-decoder token-based Transformer to input a (possibly multidimensional) time series trajectory and output a symbolic expression representing the series. Due to the lack of viable training datasets, the authors needed to generate their own synthetic dataset from randomly sampled functions.

Experiments are conducted, varying (noisy, noiseless) x (different dimensions):
* Train on synthetic dataset, test on in-distribution "test" portion of synthetic dataset
* Train on synthetic dataset, test on out-of-distribution ODEs

with comprehensive baselines (AutoML-Zero, linear regression, ElasticNet, etc.) to assess the strengths and limitations of the model.

Overall the paper is very well-written, very easy-to-understand, and the method is quite general, especially due to the ability to handle non-contiguous points in time over multivariate time series. The authors responded well to reviewers' claims of novelty + provided additional experiments dealing with e.g. missing data / irregular subsampling.

This paper is a clear accept.

**Justification For Why Not Higher Score:**

The novelty points raised by the reviewers against (https://arxiv.org/abs/2307.12617) and (https://proceedings.mlr.press/v162/d-ascoli22a.html) are still somewhat valid.

Modelling time-series data with Transformers (even in token-space) is not very new, and the paper's primary additional contributions are (1) extension to multivariate data + (2) Allowing non-contiguous time series.

**Justification For Why Not Lower Score:**

This paper is incredibly well-written and has potentially many implications and possible applications for the future, e.g.

* Sciences (Physics + Biology): General dynamical systems modeling
* AutoML (Learning curve modeling + Early Stopping)
* Language Modelling: LLM Reasoning across temporal data

This is by no means a borderline paper, and was already a clear accept. Thus I recommend at moving above the normal bar for acceptance, i.e. spotlight.

---

### Decision · Program_Chairs · 2024-01-16

Accept (spotlight)